# Supervised Disentanglement Under Hidden Correlations

## Abstract

Disentangled representation learning (DRL) methods are often leveraged to improve the generalization of representations. Recent DRL methods have tried to handle attribute correlations by enforcing conditional independence based on attributes. However, the complex multi-modal data distributions and hidden correlations under attributes remain unexplored. Existing methods are theoretically shown to cause the loss of mode information under such hidden correlations. To solve this problem, we propose Supervised Disentanglement under Hidden Correlations (SD-HC), which discovers data modes under certain attributes and minimizes mode-based conditional mutual information to achieve disentanglement. Theoretically, we prove that SD-HC is sufficient for disentanglement under hidden correlations, preserving mode information and attribute information. Empirically, extensive experiments on one toy dataset and five real-world datasets demonstrate improved generalization against the state-of-the-art baselines. Codes are available at anonymous Github https://anonymous.4open.science/r/SD-HC.

## 1 Introduction

Disentangled representation learning (DRL) aims to encode one single data attribute in each representation subspace, which holds great promise in enhancing generalization to unseen scenarios (Matthes et al., 2023; Qian et al., 2021), enabling controllable generative modeling (Yuan et al., 2021), and improving fairness (Locatello et al., 2019a). In the supervised setting, each representation subspace is learned under the label supervision of its corresponding attribute, while being disentangled from other attributes.

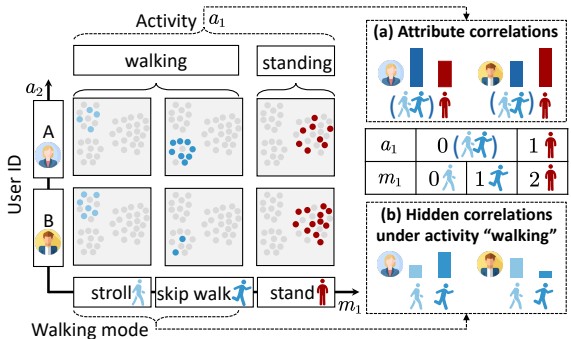

Figure 1: Correlated human activity data. The distributions of (a) **"walking" / "standing"**, and (b) **"stroll" / "skip walk"** under "walking" differ between users, exhibiting correlations.

Supervised DRL methods typically assume independence between attributes. In addition to supervised prediction, mutual information (MI) minimization (Kwon et al., 2020; Yuan et al., 2021; Su et al., 2022) is commonly adopted to enforce independence between the representations of different attributes and achieve disentanglement. The independence assumption is often violated in real-world data, where correlations are prevalent. Taking human activities as an example, different users have different behavior patterns, and each user tends to engage in some activities more frequently than others, exhibiting correlations between activity and user identity (ID) attributes, as shown in Figure 1a. For correlated attributes, enforcing representation independence causes at least one subspace to lose attribute-related information (Funke et al., 2022).

To disentangle correlated attributes, attribute-based conditional mutual information minimization (A-CMI) (Funke et al., 2022) enforces conditional representation independence that preserves attribute-related information. However, when a certain attribute takes a value, underlying variations related to this attribute may lead to complex multi-modal data distributions rather than simple uni-modal

data distributions. The mode under this value of this attribute may be correlated with other attributes. Continuing with the human activity example, when activity attribute takes the value "walking", variations in pace, stride, and posture may lead to different walking modes, the casual "stroll" and energetic "skip walk"; different users have more subtle differences in their behavior patterns, exhibiting correlations between walking mode and user ID attribute, as shown in Figure 1b. In this case, A-CMI may cause the loss of mode information (as proved in Proposition 1), which is important for attribute prediction with multi-modality (Nie et al., 2020; Sugiyama, 2021; Li et al., 2017). For example, in human activity recognition, losing the information about walking modes might lead to the confusion between "skip walk" and another activity "climbing down", while encoding mode information can better distinguish these similar activities.

To address the above problem, we propose Supervised Disentanglement under Hidden Correlations (SD-HC). Instead of focusing on attribute correlations as existing works, we delve into the complex data distributions and hidden correlations under certain attributes. Our contributions are:

- We introduce a novel supervised DRL paradigm named SD-HC, which discovers data modes under certain attributes and disentangles these attributes with mode-based CMI minimization. Under hidden correlations, SD-HC can preserve mode information that current methods tend to lose.
- We theoretically prove that mode-based CMI minimization is the *necessary and sufficient condition* for supervised disentanglement under both hidden correlations and attribute correlations. This result can be extended to show that CMI minimization can achieve disentanglement under correlations in general, establishing the first *sufficient condition* for disentanglement under correlations.
- We extensively evaluate SD-HC on five real-world datasets, which demonstrates that SD-HC outperforms the state-of-the-art DRL methods for attribute prediction on out-of-distribution data and data under train-test correlation shifts. We conduct comprehensive investigations on toy data and real-world data regarding the behavior of different methods, the impact of train correlation strength, noise level, train-test correlation shifts, and the learned representations, which demonstrate the superiority of SD-HC under various circumstances.

## 2 RELATED WORK

**Disentanged Representation Learning** DRL methods can be roughly divided into unsupervised, weakly-supervised, and supervised DRL. Unsupervised DRL learns independent representation dimensions that each correspond to an unknown attribute by self-supervised tasks, e.g., self-reconstruction in variational auto-encoders (VAEs) (Higgins et al., 2016; Kim & Mnih, 2018; Chen et al., 2018) or contrastive learning (Zimmermann et al., 2021; Matthes et al., 2023). Yet, the feasibility of purely unsupervised disentanglement has been questioned (Locatello et al., 2019b), which prompts DRL with weak supervision (Shu et al., 2020), e.g., similarity (Chen & Batmanghelich, 2020) or grouping information (Bouchacourt et al., 2018). In contrast, supervised DRL learns individual multi-dimensional representation subspaces that each encode an attribute under label supervision (Qian et al., 2021; Yuan et al., 2021). Generally, DRL methods assume attribute independence and enforce representation independence between different attributes as a means of disentanglement. In particular, supervised DRL usually minimizes the MI between attribute representations (Kwon et al., 2020; Yuan et al., 2021; Su et al., 2022), minimizes the Maximum Mean Discrepancy (MMD) between representation distributions (Li et al., 2018; Lin et al., 2020), or makes one attribute unpredictable from the representations of another by adversarial training (Qian et al., 2021; Li et al., 2022; Lee et al., 2021). Our work falls under supervised DRL.

**Disentanglement Under Attribute Correlations** Recent works have revealed that independence assumption-based DRL fails on correlated attributes, where independence constraints cause entanglement for unsupervised DRL (i.e., one dimension encodes two or more correlated attributes) (Träuble et al., 2021) or hurt the predictive ability of representations for supervised DRL (Funke et al., 2022). To disentangle correlated attributes for unsupervised DRL, Träuble et al. (Träuble et al., 2021) and Dittadi et al. (Dittadi et al., 2021) add weak supervision or a few labels to correct the model. Differently, Wang et al. (Wang & Jordan, 2021) and Roth et al. (Roth et al., 2023) relax independence constraints with Hausdorff distance to encourage only factorized supports instead of

factorized distributions. These methods can somewhat alleviate entanglement but do not guarantee disentanglement theoretically (Funke et al., 2022; Wang & Jordan, 2021).

More recent works have introduced conditional independence constraints to disentangle correlated attributes. For supervised DRL, Funke et al. (Funke et al., 2022) introduces attribute-based CMI minimization (A-CMI). For each attribute, A-CMI minimizes the MI conditioned on this attribute between its representation and the joint representations of all other attributes. A-CMI is proved to be the *necessary* condition for disentanglement under attribute correlations, whereas unconditional MI is proved to fail. For unsupervised DRL in reinforcement learning, Dunion et al. (Dunion et al., 2023) follow A-CMI, but condition on history information to make up for the unknown current values.

To the best of our knowledge, existing works have only established *necessary* conditions for disentangling correlated attributes (Wang & Jordan, 2021; Funke et al., 2022), and the sufficiency of CMI has only been validated on linear regression examples rather than proved theoretically. We are the first to give *sufficient* conditions for disentanglement under correlations, theoretically proving that CMI minimization can achieve disentanglement. Our results hold under various cases, including multiple attributes under attribute correlations and hidden correlations.

## 3 Disentanglement under Hidden Correlations

### 3.1 Problem Formulation

**Data Generation Process.** We assume that data are generated from the causal process in Definition 1 and Figure 2, which mainly relies on the *independent mechanism assumption* (Schölkopf et al., 2012) that attributes are casually independent, i.e., each attribute arises on its own, allowing changes or interventions on one attribute without affecting others. Still, confounding may exist.

**Definition 1.** (Disentangled Causal Process). *Consider a causal generative model $p(\boldsymbol{x}|\boldsymbol{a})$ for data $\boldsymbol{x}$ with $K$ attributes $\boldsymbol{a} = (a_1, a_2, ..., a_K)$. A certain attribute $a_k$ is associated with a categorical mode variable $m_k$. Attributes $\boldsymbol{a}$ could be influenced by $L$ confounders $\boldsymbol{c}^a = (c_1^a, ..., c_L^a)$. Conditioned on $a_k$, mode variable $m_k$ and other attributes $\boldsymbol{a}_{-k}$ could be influenced by $Q$ confounders $\boldsymbol{c}^m = (c_1^m, ..., c_Q^m)$. This causal model is called disentangled if and only if it can be described by a structural causal model (SCM) (Pearl, 2009) of the form:*

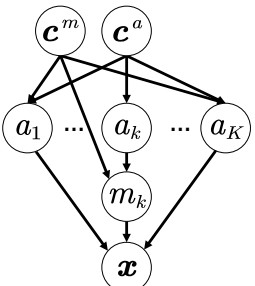

$$\boldsymbol{c}^a \leftarrow \boldsymbol{n}^{ca}, \boldsymbol{c}^m \leftarrow \boldsymbol{n}^{cm}$$
$$a_k \leftarrow h_k^a(S_k^a, \boldsymbol{n}_k^a), S_k^a \subset \{c_1^a, ..., c_L^a\}, k \in \{1, ..., K\}$$
$$a_j \leftarrow h_j^a(S_j^a, S_j^m, \boldsymbol{n}_j^a), S_j^a \subset \{c_1^a, ..., c_L^a\}, S_j^m \subset \{c_1^m, ..., c_Q^m\}, j \neq k$$
$$m_k \leftarrow h^m(a_k, \boldsymbol{c}^m, \boldsymbol{n}^m)$$
$$\boldsymbol{x} \leftarrow g(\boldsymbol{a}_{-k}, m_k, \boldsymbol{n}^x)$$

(1)

Figure 2: Causal graph of data generation with multi-modality and hidden correlations under a certain $a_k$.

with functions $g, h_i^a, h^m$, jointly independent noise variables $\boldsymbol{n}^{ca}, \boldsymbol{n}^{cm}, \boldsymbol{n}_i^a, \boldsymbol{n}^m, \boldsymbol{n}^x$, and confounder subsets $S_i^a, S_j^m$, for $i = 1, ..., K, j = 1, ..., k-1, k+1, ..., K$. $-k$ denotes the set of attribute indices $\{j\}_{j \neq k}$. Note that $\forall i \neq i', a_i \not\rightarrow a_{i'}$.

**Correlations.** We denote mutual information (MI) and entropy function as $I(\cdot\,;\cdot)$ and $H(\cdot)$, respectively. The MI between attributes measures their correlations, e.g., $I(a_i; a_{i'}), i \neq i'$, while the MI between a representation and an attribute measures the amount of information the representation contains about the attribute, e.g., $I(\boldsymbol{z}_i; a_{i'})$. We denote attribute correlations as $I(a_i; a_{i'})$ and hidden correlations as $I(m_k; a_{-k}|a_k)$, which are induced by confounders $\boldsymbol{c}^a$ and $\boldsymbol{c}^m$, respectively.

**Multi-Modality and Hidden Correlations.** Under some value $\alpha$ of $a_k$, the data distribution is assumed to be multi-modal due to underlying variations related to this attribute, i.e., $p(\boldsymbol{x}|a_k = \alpha)$ is a mixture model, e.g., Gaussian mixture model, and a mode corresponds to a component of the mixture. The modes under different attribute values are labeled altogether to formulate the categorical variable $m_1$, e.g., for $a_k$ with 2 values ($|\mathcal{A}_k| = 2$) and 3 modes under each value, the 6 modes in total will be labeled from 0 to 5 to formulate $m_1$. The modes under $a_k = \alpha$ may be correlated with other attributes $a_{-k}$, i.e., $I(m_k; a_{-k}|a_k = \alpha) > 0$. Hidden correlations are defined as the expectation over different attribute values, i.e., $I(m_k; a_{-k}|a_k) = \sum_{\alpha \in \mathcal{A}_k} p_{a_k}(a_k = \alpha)I(m_k; a_{-k}|a_k = \alpha)$.

**Intuitive Example.** Figure 1 illustrates hidden correlations in the realistic application of human activity recognition: Under different values of activity attribute $a_1$, two walking modes and one standing mode are labeled altogether to formulate variable $m_1$, i.e., $m_1 = 0, 1, 2$ indicates "stroll", "skip walk", and "stand", respectively; The modes "stroll" and "skip walk" under "walking" activity might be correlated with user ID attribute $a_2$, i.e., $I(m_1; a_2 | a_1 = 0) > 0$, where $a_1 = 0$ indicates "walking" activity and $m_1 | a_1 = 0$ indicates walking modes $m_1 = 0, 1$. For activity recognition, the goal is to learn disentangled activity representations that fully capture the activity and its modes, while remaining unaffected by personalized user patterns.

## 3.2 THE DEFINITIONS OF DISENTANGLED REPRESENTATIONS

The goal of supervised DRL is to learn disentangled representations $z_i$ for each attribute $a_i$ by a mapping $f(x) = z = (z_1, z_2, ..., z_K), z_i \in \mathbb{R}^D, i = 1, ..., K$. Disentangled $z_i$ should (1) contain all information about $a_i$ (***Informativeness***), which includes mode information for attributes with multi-modality, and (2) reflect the *causal independence* between attributes, such that external interventions by changing another attribute $a_j (i \neq j)$ alone should not affect $z_i$ (***Independence***), which is formalized in Definition 2 following (Wang & Jordan, 2021; Suter et al., 2019).

**Definition 2.** (Disentangled Representation). *Representation $z$ is disentangled, if for $i = 1, ..., K$:*

$$p(z_i | \mathrm{do}(a_{-i})) = p(z_i) \tag{2}$$

where $-i$ indicates the set of attribute indices $\{j\}_{j \neq i}$, $a_{-i}$ indicates the joint variable of $\{a_j\}_{j \neq i}$, and $\mathrm{do}(\cdot)$ operation sets the values of some attributes by external intervention and leaves other attributes unchanged. Such external intervention is isolated from the causal effects within the causal process.

## 3.3 THEORETICAL GUARANTEES FOR DISENTANGLING WITH MODE-BASED CMI MINIMIZATION

We focus on the disentanglement of a certain attribute with multi-modality, and show that under the *independent mechanism assumption*, mode-based CMI minimization is the necessary and sufficient condition for supervised disentanglement under hidden correlations and attribute correlations, while A-CMI fails under hidden correlations. For simplicity, we take $K = 2$ as an example, with $a_1$ exhibiting multi-modality. Then, the results are generalized to multiple attributes and simple cases.

**The Necessary Condition for Disentanglement.** Based on the data generation process of Definition 1, we build the causal graphs of the true latent representations (denoted as $z_k^l$), which are only causally dependent on the corresponding attribute or mode, and are inherently disentangled.

Since the ideally disentangled $z_k$ should capture the true latent $z_k^l$ and retain its properties, we find conditional independence between the true latent representations as the necessary condition for disentanglement. As stated by the causal graph theorems in Appendix C, two variables $X, Y$ are conditionally independent given a variable that blocks all *backdoor paths* between them, i.e., the paths that flow backward from $X$ or $Y$. In Figure 3a, we consider only attribute correlations as A-CMI, where $a_1$ blocks the only *backdoor path* between $z_1^l$ and $z_2^l$. In comparison, we consider hidden correlations and potential attribute correlations in Figure 3b, where $m_1$ blocks all *backdoor paths* whether attribute correlations exist or not, yet $a_1$ fails to block the path containing $c^m$. This means that under hidden correlations and potential attribute correlations, disentangled representations should retain the conditional independence of the true latent representations as:

$$I(z_1^l; z_2^l | m_1) = 0 \quad \Rightarrow \quad I(z_1; z_2 | m_1) = 0 \tag{3}$$

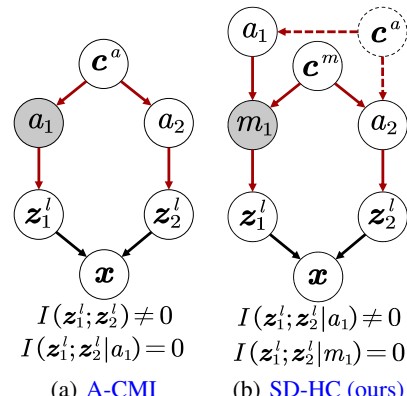

$$I(z_1^l; z_2^l) \neq 0$$
$$I(z_1^l; z_2^l | a_1) = 0$$
(a) A-CMI

$$I(z_1^l; z_2^l | a_1) \neq 0$$
$$I(z_1^l; z_2^l | m_1) = 0$$
(b) SD-HC (ours)

Figure 3: Causal graphs of the true latent representations. **Red** arrows indicate the *backdoor paths* between $z_1^l$ and $z_2^l$. The dashed circle and arrows indicate that attribute correlations may or may not exist.

**A-CMI Fails Under Hidden Correlations.** As shown in Figure 3b, disentangled representations $z_1$ and $z_2$ are not conditionally independent given $a_1$ under hidden correlations. We further show that

enforcing such conditional independence could hurt the predictive ability of representations, which is formalized in Proposition 1 and proved in Appendix B.2.

**Proposition 1.** *For representations $z_1, z_2$ of $m_1, a_2$, respectively, if $I(m_1; a_2|a_1) > 0$, then enforcing $I(z_1; z_2|a_1) = 0$ leads to at least one of $I(z_1; m_1) < H(m_1)$ and $I(z_2; a_2) < H(a_2)$.*

where $I(z_1; m_1) < H(m_1)$ indicates that $z_1$ fails to contain mode information for predicting $a_1$, and $I(z_2; a_2) < H(a_2)$ indicates that $z_2$ fails to contain attribute-related information for predicting $a_2$. Either way, attribute-based CMI minimization $I(z_1; z_2|a_1) = 0$ hurts the predictive ability of representations under hidden correlations. This is an extension of Proposition 3.1 in (Funke et al., 2022), which proves that unconditional MI minimization fails under attribute correlations.

**The Sufficient Condition for Disentanglement.** Since inappropriate independence constraints could hurt the predictive ability of representations, the key to disentanglement is to find suitable independence constraints. We show that mode-based CMI minimization (Equation 3) is sufficient for supervised disentanglement under the *independent mechanism assumption* with various correlations.

For the two criteria of disentanglement: (1) ***Informativeness*** requires $I(z_1; a_1) = H(a_1)$ and $I(z_1; m_1) = H(m_1)$, which can be achieved by cross-entropy minimization (Boudiaf et al., 2020). Since mode-based CMI minimization has been proven necessary for disentanglement, it preserves the predictive ability of representations. (2) ***Independence*** is a bit tricky, as the impact of external interventions cannot be directly evaluated (Wang & Jordan, 2021). We prove that mode-based CMI minimization ensures representations are conditionally independent of other attributes (Proposition 2, Appendix B.3), and then prove that under the *independent mechanism assumption*, this conditional independence yields disentanglement in the sense of Definition 2 (Proposition 3, Appendix B.4).

**Proposition 2.** *For representations $z_1, z_2$ of $m_1, a_2$, respectively, if $I(z_1; m_1) = H(m_1)$, $I(z_2; a_2) = H(a_2)$, and $I(z_1; z_2|m_1) = 0$, then $I(z_1; a_2) = I(m_1; a_2)$ and $I(z_1; a_2|m_1) = 0$.*

where $I(m_1; a_2)$ is denoted as the *total hidden correlations* between $m_1$ and $a_2$. As proved in Appendix B.1, total hidden correlations are the sum of attribute correlations and hidden correlations, i.e., $I(m_1; a_2) = I(a_1; a_2) + I(m_1; a_2|a_1)$. Thereby, $I(z_1; a_2) = I(m_1; a_2)$ shows that $z_1$ contains information about $a_2$ only if it is induced by correlations regarding its attribute or mode. Furthermore, $I(z_1; a_2|m_1) = 0$ shows that $z_1$ contains no additional information about $a_2$ knowing its mode.

**Proposition 3.** *Under the data generation assumption of Definition 1 ($K = 2$, $k = 1$) with independent mechanisms, if $I(z_1; a_2|m_1) = 0$ for representation $z_1$, then $p(z_1|\mathrm{do}(a_2)) = p(z_1)$.*

This is proved by do-calculus (Pearl, 2009), linking to Definition 2 and completing our proof.

**Generalization to Multiple Attributes and Simple Cases.** Our theoretical results naturally generalize to (1) $K > 2$, where the extension mainly involves replacing single variables $a_2, z_2$ with joint variables $a_{-k}, z_{-k}$, as discussed in Appendix B.5; (2) simple uni-modal data with attribute correlations, where the number of modes under each attribute value reduces to 1, and mode-based CMI degrades to attribute-based CMI; and (3) simple uncorrelated data, where confounding can be neglected, and mode-based CMI performs similarly to attribute-based CMI, as shown in Figure 8cd.

**Theoretical Contributions.** We prove the sufficiency of CMI for disentanglement, while the work of A-CMI only validates CMI on linear regression examples without formal proof. *This is the first attempt to establish sufficient conditions for disentanglement under correlations*, unlike necessary conditions before (Wang & Jordan, 2021; Funke et al., 2022). Our results generalize to multiple attributes, various correlation types, and simple uni-modal and uncorrelated data, showing that one independence constraint is sufficient for the supervised disentanglement of one representation $z_k$. Formally, under the *independent mechanism assumption* in Definition 1, given the label supervision of all attributes and modes, when the supervised losses on all attributes and modes are optimized, and the CMI of $z_k$ ($I(z_k; z_{-k}|m_k)$ for multi-modal or $I(z_k; z_{-k}|a_k)$ for uni-modal cases) is minimized, the learned $z_k$ is disentangled in the sense of Definition 2, as elaborated in Appendix B.5.

## 4 METHOD

**Framework.** We show the framework of SD-HC for disentangling a certain attribute $a_k$ with hidden correlations in Figure 4, which consists of encoder $F$ for learning representations $F(x) = z = (z_1, z_2, ..., z_K), z_i \in \mathbb{R}^D, i = 1, ..., K$, predictors $\{C_i\}_{i=1}^K$ for predicting each attribute, predictor

Figure 4: Framework of SD-HC for disentangling a certain $a_k, k \in \{1, ..., K\}$ with multi-modality.

$C_k^m$ for predicting mode $m_k$, and discriminator $D_k$ for minimizing mode-based CMI. SD-HC is architecture-agnostic and can be used in various applications.

The framework can be expanded to disentangle multiple attributes by adding one independence constraint to disentangle each attribute. The form of independence constraints depends on the correlation types, i.e., minimizing attribute-based CMI under attribute correlations or mode-based CMI under hidden correlations. Supervised constraints $I(\boldsymbol{z}_i; a_i) = H(a_i)$ are always required for $i = 1, ..., K$ with one additional constraint $I(\boldsymbol{z}_i; m_i) = H(m_i)$ for each attribute $a_i$ with multi-modality. For additional constraints, discriminators and mode predictors should be added accordingly.

**Mode Label Estimation.** We assume the attribute labels are known, while the number of modes and the mode labels are unknown. To estimate mode labels for $a_k$, we perform clustering on the representations of a pre-trained encoder, which is trained with the supervised loss of $a_k$. Specifically, given the number of modes $N_m$, clustering is performed under each value of $a_k$, then the discovered modes under different values of $a_k$ are labeled altogether to formulate $m_k$. We adopt k-means as the clustering method, which works well across our experiments. The numbers of modes $N_m$ under different values of $a_k$ are set to be equal and tuned as a hyper-parameter. We also provide practical guidance for different scenarios in Appendix I, including the alternative clustering methods and mode number estimation methods for reducing the computational costs of hyper-parameter tuning.

**Losses.** The losses are strictly designed according to the sufficient conditions for disentanglement. As commonly done in adversarial training (Chen et al., 2023), optimizations w.r.t. different losses are performed alternatively. The detailed training process is given in Appendix E.

(1) For supervised learning, attribute and mode prediction losses $\mathcal{L}_{ac}, \mathcal{L}_{mc}$ are formulated as:

$$\mathcal{L}_{ac} = \mathbb{E}_{\boldsymbol{x}}[\textstyle\sum_{k=1}^{K} l_{ce}(C_k(F_k(\boldsymbol{x})), a_k)] \ , \ \mathcal{L}_{mc} = \mathbb{E}_{\boldsymbol{x}}[l_{ce}(C_k^m(F_k(\boldsymbol{x})), m_k)] \tag{4}$$

$$\mathcal{L}_c = \mathcal{L}_{ac} + w_m \cdot \mathcal{L}_{mc} \tag{5}$$

where $w_m$ is the weight of mode prediction loss, and $l_{ce}(\cdot)$ denotes cross entropy function.

(2) Since $I(\boldsymbol{z}_k; \boldsymbol{z}_{-k}|m_k) = 0$ if and only if $p(\boldsymbol{z}_k, \boldsymbol{z}_{-k}|m_k) = p(\boldsymbol{z}_k|m_k)p(\boldsymbol{z}_{-k}|m_k)$, we minimize CMI by matching the joint distribution $p(\boldsymbol{z}_k, \boldsymbol{z}_{-k}|m_k)$ with the marginal distribution $p(\boldsymbol{z}_k|m_k)p(\boldsymbol{z}_{-k}|m_k)$ with adversarial training (Belghazi et al., 2018). To sample from the two distributions, we loop over the values of mode labels $\mu \in \{0, ..., N_m * |\mathcal{A}_k| - 1\}$. For each value $\mu$, we select the representations in the mini-batch with label $m_k = \mu$. The samples from the joint distribution are obtained by concatenating the selected representations, and those from the marginal distribution are obtained by shuffling the selected $\boldsymbol{z}_{-k}$ jointly then concatenating them with the selected $\boldsymbol{z}_k$. Jensen-Shannon Divergence is used to measure the discrepancy between the two distributions for stability (Hjelm et al., 2019). Discrimination loss $\mathcal{L}_d$ is formulated as follows, where $l_{bce}(\cdot)$ denotes binary cross entropy function:

$$\mathcal{L}_d = \mathbb{E}_{m_k}[\mathbb{E}_{(\boldsymbol{z}_k, \boldsymbol{z}_{-k})|m_k}[l_{bce}(D(\boldsymbol{z}_k, \boldsymbol{z}_{-k}, m_k), 1)] + \mathbb{E}_{\boldsymbol{z}_k|m_k, \boldsymbol{z}_{-k}|m_k}[l_{bce}(D(\boldsymbol{z}_k, \boldsymbol{z}_{-k}, m_k), 0)]] \tag{6}$$

## 5 EXPERIMENTS

### 5.1 EXPERIMENTAL SETTINGS

**Datasets.** We evaluate on one toy dataset and five real-world datasets in various applications, i.e., digit recognition, wearable human activity recognition (WHAR), and machine fault diagnosis. The datasets are described as follows. See Appendix F for more details.

(1) **Toy** dataset is constructed as two-dimensional data with two binary attributes $a_1, a_2$, as shown in Figure 5a. $a_1$ has three modes under each attribute value. Data are generated through linearly mapping $m_1$ and $a_2$ to two-dimensional spaces and adding noises with noise level $\sigma$.

(2) Colored MNIST (**CMNIST**) is constructed from MNIST (LeCun et al., 1998), as shown in Figure 5b. Parity check identifies whether a digit is even or odd with multiple digits under each parity value. Accordingly, $a_1$ represents parity, $m_1$ represents the digits, and $a_2$ represents the color of digits, which is often correlated with digits, e.g., a player's jersey number may be associated with a specific color in sports. Noises are introduced to both digits and colors, and digit noises are generated as occlusions with occlusion ratio as the noise level $\sigma$ (Chai et al., 2021).

(3) **UCI-HAR** (Anguita et al., 2013), **RealWorld** (Sztyler & Stuckenschmidt, 2016), **HHAR** (Stisen et al., 2015) record wearable sensor data, from which WHAR identifies activities with variations under each activity. Accordingly, $a_1$ represents activity, $m_1$ represents unknown activity modes, and $a_2$ represents user ID, which is often correlated with activity due to personal behavior patterns.

(4) **MFD** (Lessmeier et al., 2016) record sensor data from bearing machines, from which machine fault diagnosis identifies machine fault types with variations under each fault type, e.g., different forms of damages. Accordingly, $a_1$ represents fault type, $m_1$ represents unknown modes of fault types, and $a_2$ represents operating conditions, which could be correlated with machine faults.

**Evaluation Protocols.** *On toy and CMNIST datasets, correlations are introduced by sampling* (Roth et al., 2023). As illustrated in Figure 6, we train on correlated data and evaluate on 3 test sets, namely test 1 under the same correlations, test 2 without correlations, and test 3 with anticorrelations. The train-test correlation shift increases from test 1 to 3. For comparison with baselines and variants on CMNIST, we train under both attribute correlations and hidden correlations, and report the results on test 3. The complete results on all test sets are in Appendix J. For additional analysis, we train under $cor_h = I(m_1; a_2|a_1) > 0$ to focus on hidden correlations. *On other datasets, we investigate DRL under natural correlations*, where the number of modes and mode labels are unknown. Leave-one-group-out validation is performed, where users and operating

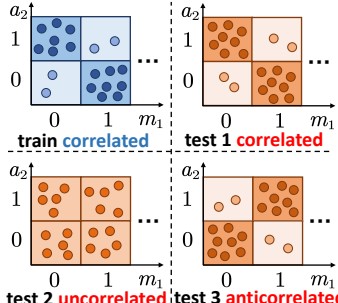

Figure 6: Train-test setup.

conditions in the test group are unseen during training, formulating out-of-distribution (OOD) tasks.

On each dataset, we focus on disentangling one attribute with multi-modality as mentioned before. Accuracy (Acc.) and macro F1 score (Mac. F1) are used as performance metrics for attribute prediction. These metrics can reflect the disentanglement of representations under correlation shifts and OOD tasks (Funke et al., 2022; Dittadi et al., 2021), while common disentanglement metrics are not suitable under correlations (Locatello et al., 2020). Each experiment is repeated using 5 varying random seeds, with the mean and standard deviation reported. See details in Appendix F.

**Baselines and Implementations.** We compare SD-HC with typical DRL methods (**MMD** (Lin et al., 2020), **DTS** (Li et al., 2022), **IDE-VC** (Yuan et al., 2021), and **MI** (Cheng et al., 2022)), and the state-of-the-art DRL methods under correlations (**A-CMI** (Funke et al., 2022) and **HFS** (Oublal et al., 2024)). For reference, we also include the base method trained on supervised prediction losses only, which is denoted as **BASE**, and a variant of our method that uses ground truth mode labels

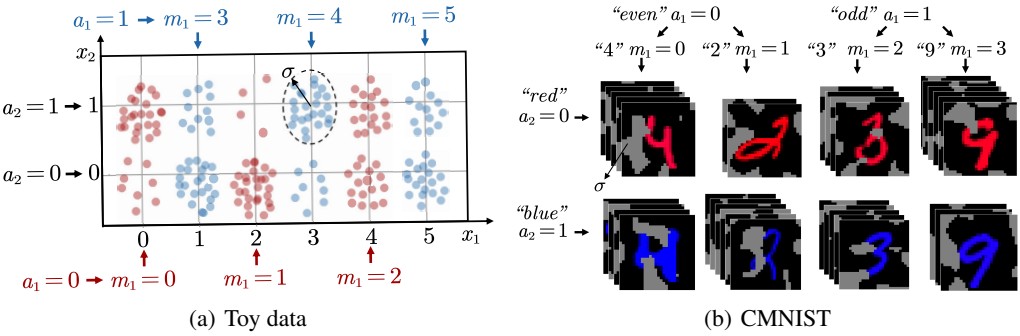

(a) Toy data          (b) CMNIST

Figure 5: Data construction of toy dataset and CMNIST with noise level $\sigma$.

Table 1: Comparison with baselines (mean±std). "*" indicates that SD-HC is statistically superior to the baselines by pairwise t-test at a 95% significance level. The results of the best methods are **bold**. The results of the runner-up methods are underlined, over which the improvement is calculated.

| Method | CMNIST | | UCI-HAR | | RealWorld | | HHAR | | MFD | |
|---|---|---|---|---|---|---|---|---|---|---|
| | Acc. | Mac. | Acc. | Mac. F1 | Acc. | Mac. F1 | Acc. | Mac. F1 | Acc. | Mac. F1 |
| BASE | $\underline{0.768}_{\pm0.008}$* | $\underline{0.768}_{\pm0.008}$* | $0.712_{\pm0.028}$* | $0.697_{\pm0.036}$* | $0.646_{\pm0.014}$* | $0.654_{\pm0.014}$* | $0.808_{\pm0.016}$* | $0.809_{\pm0.020}$* | $0.727_{\pm0.016}$* | $0.763_{\pm0.009}$* |
| MMD | $0.582_{\pm0.069}$* | $0.528_{\pm0.117}$* | $0.703_{\pm0.037}$* | $0.662_{\pm0.035}$* | $\underline{0.660}_{\pm0.019}$* | $0.652_{\pm0.023}$* | $\underline{0.809}_{\pm0.012}$* | $0.805_{\pm0.017}$* | $0.782_{\pm0.019}$* | $0.791_{\pm0.016}$* |
| DTS | $0.615_{\pm0.022}$* | $0.615_{\pm0.022}$* | $0.728_{\pm0.033}$* | $0.701_{\pm0.026}$* | $0.644_{\pm0.023}$* | $0.649_{\pm0.015}$* | $0.798_{\pm0.024}$* | $0.797_{\pm0.017}$* | $0.670_{\pm0.022}$* | $0.674_{\pm0.015}$* |
| IDE-VC | $0.539_{\pm0.022}$* | $0.533_{\pm0.027}$* | $0.736_{\pm0.031}$* | $0.732_{\pm0.034}$* | $0.652_{\pm0.013}$* | $0.650_{\pm0.017}$* | $0.807_{\pm0.020}$* | $0.806_{\pm0.014}$* | $0.741_{\pm0.018}$* | $0.763_{\pm0.011}$* |
| MI | $0.611_{\pm0.039}$* | $0.600_{\pm0.044}$* | $\underline{0.749}_{\pm0.021}$* | $\underline{0.745}_{\pm0.027}$* | $\underline{0.660}_{\pm0.018}$* | $\underline{0.655}_{\pm0.016}$* | $\underline{0.809}_{\pm0.017}$* | $\underline{0.807}_{\pm0.021}$* | $0.763_{\pm0.012}$* | $0.776_{\pm0.016}$* |
| A-CMI | $0.611_{\pm0.039}$* | $0.600_{\pm0.044}$* | $0.714_{\pm0.034}$* | $0.700_{\pm0.030}$* | $0.654_{\pm0.015}$* | $\underline{0.655}_{\pm0.012}$* | $0.802_{\pm0.018}$* | $0.803_{\pm0.023}$* | $\underline{0.788}_{\pm0.014}$* | $\underline{0.798}_{\pm0.007}$* |
| HFS | $0.635_{\pm0.008}$* | $0.631_{\pm0.008}$* | $0.671_{\pm0.035}$* | $0.651_{\pm0.040}$* | $0.489_{\pm0.018}$* | $0.398_{\pm0.015}$* | $0.782_{\pm0.012}$* | $0.783_{\pm0.015}$* | $0.754_{\pm0.017}$* | $0.710_{\pm0.013}$* |
| SD-HC | $\mathbf{0.829}_{\pm0.011}$ | $\mathbf{0.829}_{\pm0.008}$ | $\mathbf{0.830}_{\pm0.03}$ | $\mathbf{0.833}_{\pm0.036}$ | $\mathbf{0.698}_{\pm0.019}$ | $\mathbf{0.699}_{\pm0.014}$ | $\mathbf{0.845}_{\pm0.023}$ | $\mathbf{0.842}_{\pm0.015}$ | $\mathbf{0.825}_{\pm0.020}$ | $\mathbf{0.825}_{\pm0.015}$ |
| **Improvement** | ↑6.1 % | ↑6.1 % | ↑7.0 % | ↑7.4 % | ↑4.0 % | ↑4.4 % | ↑3.6 % | ↑3.5 % | ↑3.7 % | ↑3.2 % |

Table 2: Comparison with variants (mean±std). The notations are the same as Table 1.

| Method | CMNIST | | UCI-HAR | | RealWorld | | HHAR | | MFD | |
|---|---|---|---|---|---|---|---|---|---|---|
| | Acc. | Mac. F1 | Acc. | Mac. F1 | Acc. | Mac. F1 | Acc. | Mac. F1 | Acc. | Mac. F1 |
| SD-HC-A | $0.771_{\pm0.009}$* | $0.770_{\pm0.009}$* | $\underline{0.822}_{\pm0.023}$ | $0.823_{\pm0.027}$ | $0.639_{\pm0.016}$* | $0.634_{\pm0.014}$* | $0.819_{\pm0.016}$* | $0.813_{\pm0.021}$ | $0.815_{\pm0.016}$ | $0.814_{\pm0.011}$ |
| SD-HC-MG | $0.797_{\pm0.012}$* | $0.797_{\pm0.012}$* | $\underline{0.822}_{\pm0.020}$ | $0.828_{\pm0.029}$ | $\underline{0.684}_{\pm0.015}$* | $\underline{0.688}_{\pm0.020}$ | $0.806_{\pm0.017}$* | $0.802_{\pm0.023}$* | $0.803_{\pm0.015}$* | $0.804_{\pm0.018}$* |
| SD-HC-ID | $\underline{0.802}_{\pm0.015}$* | $\underline{0.802}_{\pm0.15}$* | $0.776_{\pm0.018}$ | $0.768_{\pm0.023}$ | $0.683_{\pm0.012}$* | $0.678_{\pm0.011}$* | $0.772_{\pm0.019}$* | $0.755_{\pm0.015}$* | $0.806_{\pm0.017}$* | $0.809_{\pm0.012}$ |
| SD-HC-SD | $0.783_{\pm0.010}$* | $0.783_{\pm0.010}$* | $0.774_{\pm0.018}$ | $0.768_{\pm0.018}$ | $0.662_{\pm0.013}$* | $0.666_{\pm0.018}$* | $0.810_{\pm0.024}$* | $0.812_{\pm0.018}$* | $0.792_{\pm0.018}$* | $0.792_{\pm0.013}$* |
| SD-HC-M | $0.832_{\pm0.009}$* | $0.832_{\pm0.009}$* | $0.796_{\pm0.029}$ | $0.792_{\pm0.032}$ | $0.672_{\pm0.018}$* | $0.676_{\pm0.020}$* | $\underline{0.839}_{\pm0.023}$ | $\underline{0.835}_{\pm0.017}$ | $\underline{0.817}_{\pm0.015}$ | $\underline{0.817}_{\pm0.020}$ |
| SD-HC | $\mathbf{0.829}_{\pm0.011}$ | $\mathbf{0.829}_{\pm0.008}$ | $\mathbf{0.830}_{\pm0.03}$ | $\mathbf{0.833}_{\pm0.036}$ | $\mathbf{0.698}_{\pm0.019}$ | $\mathbf{0.699}_{\pm0.014}$ | $\mathbf{0.845}_{\pm0.023}$ | $\mathbf{0.842}_{\pm0.015}$ | $\mathbf{0.825}_{\pm0.020}$ | $\mathbf{0.825}_{\pm0.015}$ |

when available, which is denoted as **SD-HC-T**. All methods are implemented using PyTorch (Paszke et al., 2019). Please see Appendix H, F, D, G for details of baselines, implementations, network architectures, and hyper-parameter tuning, respectively. Our codes are available at anonymous Github.

## 5.2 COMPARISON WITH BASELINE DRL METHODS

The comparison with baseline DRL methods is shown in Table 1, from which we observe:

(1) SD-HC consistently shows superiority over the compared baselines, outperforming the best baseline by an average of 4.86% and 4.92% in accuracy and macro F1 score, respectively. This indicates that SD-HC can better disentangle representations by improving the generalization ability while preserving the predictive ability. For introduced correlations, the large performance gain on CMNIST indicates that SD-HC is advantageous under attribute correlations and strong hidden correlations. For natural correlations, the large performance gain on UCI-HAR indicates the advantage of SD-HC on real-world data with complex multi-modality and hidden correlations.

(2) Despite considering correlations, A-CMI and HFS still fail to improve over BASE in some cases. A-CMI deals with attribute correlations, but fails under hidden correlations in losing mode information. HFS deals with correlations in general, yet its assumption of factorized support might not hold and hurt the predictive ability of representations. For example, in WHAR, HFS assumes that the probability of some user performing some activity could be low, but each user still performs each activity and each activity mode; this is often violated, as users might not perform certain activities due to personalized behavior patterns.

(3) MMD, DTS, IDE-VC, and MI fail to improve over BASE in some cases, because they do not consider correlations and might hurt the predictive ability of representations. Their performance degradation from BASE is especially severe on CMNIST under large train-test correlation shifts.

## 5.3 COMPARISON WITH VARIANTS

We compare with the following variants: **SD-HC-A** additionally minimizes attribute-based CMI for the attributes without multi-modality; **SD-HC-MG** uses Marigold (Mortensen et al., 2023) instead of k-means for clustering in high dimensional spaces; **SD-HC-ID** and **SD-HC-SD** use individual

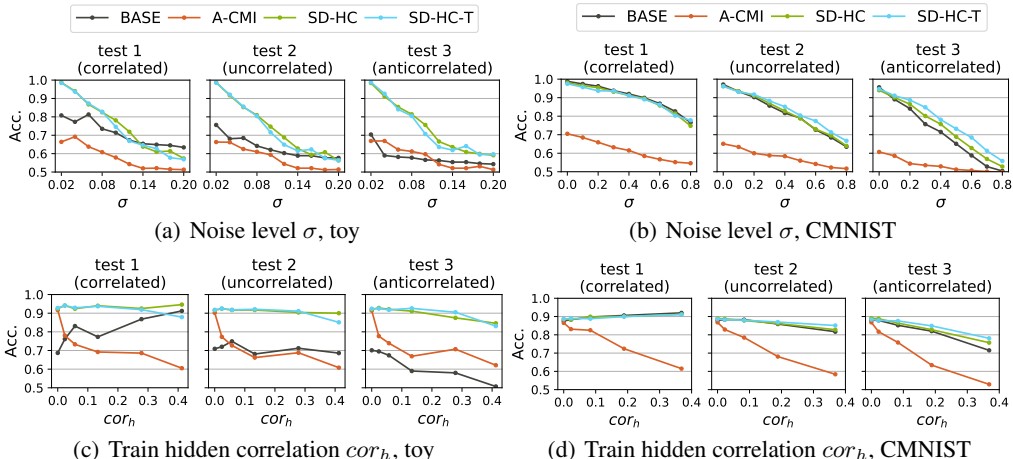

(a) Noise level $\sigma$, toy       (b) Noise level $\sigma$, CMNIST

(c) Train hidden correlation $cor_h$, toy       (d) Train hidden correlation $cor_h$, CMNIST

Figure 8: Impact of noise level and train hidden correlation on toy and CMNIST datasets.

discriminators and one shared discriminator for different modes, respectively; **SD-HC-M** minimizes mode prediction loss on some inter-mediate representations of encoder. See Appendix D for details.

The results are shown in Table 2, which shows that: (1) SD-HC-A generally does not improve SD-HC, probably because one independence constraint is sufficient for disentanglement, and additional adversarial training objectives might increase training difficulty. (2) SD-HC-MG generally does not improve SD-HC, indicating that k-means is effective for our 128-dimensional representations. Marigold could be considered as an alternative for representations of higher dimensions. (3) SD-HC-ID and SD-HC-SD consistently underperform SD-HC, indicating that proper parameter sharing between different modes in the discriminator is beneficial for mode-based CMI minimization. (4) SD-HC-M consistently underperforms SD-HC, indicating that enforcing mode and attribute prediction on the same representation space benefits the encoding of mode information.

## 5.4 METHOD INVESTIGATIONS

**Toy Decision Boundary.** On toy data, the decision boundaries of the trained predictor of $a_1$ and the prediction accuracy on the three test sets are shown in Figure 7, from which we observe:

(1) The upper right boundaries of BASE surround the clusters at $a_2 = 1$, and BASE performance decreases as the correlation shift enlarges from test 1 to 3, which indicates that without independence constraints, BASE over-encodes $a_2$ and lacks generalization ability.

(2) The decision boundaries of A-CMI span across the clusters at $a_2 = 0, 1$ without excluding either value, but fail to separate interleaving clusters at different values of $m_1$, and the performance is low but robust across 3 test sets, indicating that A-CMI does not over-encode $a_2$, but loses important mode information.

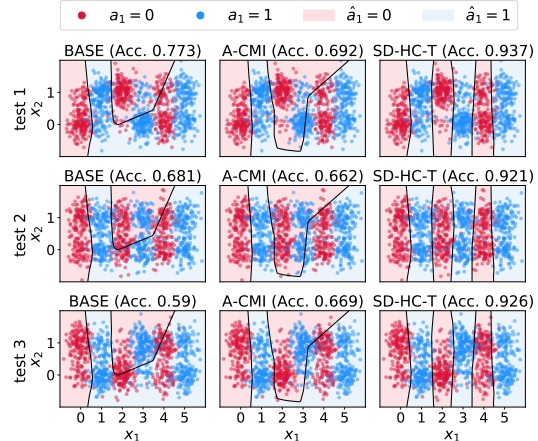

Figure 7: Decision boundary on toy data.

(3) The decision boundaries of SD-HC-T almost conform to vertical lines $x_1 = b$ that distinguish interleaving clusters, and SD-HC-T shows robustness and superiority across different correlation shifts on 3 test sets. This indicates that SD-HC-T can learn mode information about $a_1$ (***Informativeness***), and exclude irrelevant information about $a_2$ (***Independence***), achieving disentanglement.

**The Impact of Noise Level and Train Hidden Correlation.** The prediction accuracy of $a_1$ under varying noise levels and train hidden correlations is shown in Figure 8, which shows that:

(1) SD-HC generally shows superiority under varying noises and correlations due to its ability to achieve disentanglement. In addition, the comparable performance of SD-HC and SD-HC-T indicates the effectiveness of mode label estimation by k-means on synthetic and real data.

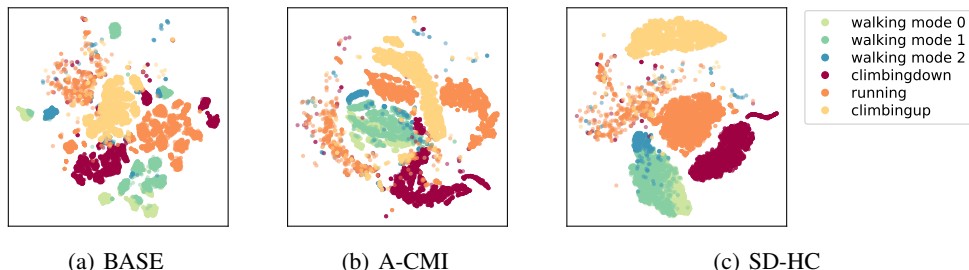

(a) BASE        (b) A-CMI        (c) SD-HC

Figure 9: Visualization of activity representation distributions on RealWorld. (a), (b), and (c) show the results of four similar activities in BASE, A-CMI, and SD-HC. Compared to SD-HC, activities "walking" and "climbing down" are confused in A-CMI for losing information about walking modes.

(2) In Figure 8a test 1, BASE performs the best at large noise levels, because BASE makes up for the noise-induced information loss by over-encoding $a_2$. In Figure 8c test 1, BASE performs the best at large train correlations, because when the correlation between $m_1$ and $a_2$ is larger, over-encoding $a_2$ leads BASE to learn more mode information and better predict $a_1$. As the correlation shift enlarges from test 1 to 3, these advantages are lost due to the lack of generalization ability.

(3) In Figure 8cd, A-CMI performs comparably to SD-HC without hidden correlations ($cor_h = 0$); yet A-CMI performance decreases as hidden correlation increases, because A-CMI does not allow representations to encode shared information induced by hidden correlations, and loses more mode information as hidden correlation increases. The behavior of A-CMI reflects the behavior of common DRL methods that overlook hidden correlations, demonstrating a broader significance.

**WHAR Visualizations.** On the training set of RealWorld, the data of estimated modes are visualized in Figure 10, which shows that the signals of the three walking modes differ in mean values and volatility, possibly due to varying paces, strides, and postures.

The activity representation distributions are visualized by t-SNE in Figure 9, which shows that: (1) **BASE** representations are separated within each activity, probably due to over-encoding user ID and learning personalized user patterns. (2) **A-CMI** representations of different walking modes and different activities are mixed, indicating that different activities are confused due to the loss of mode information. (3) **SD-HC**

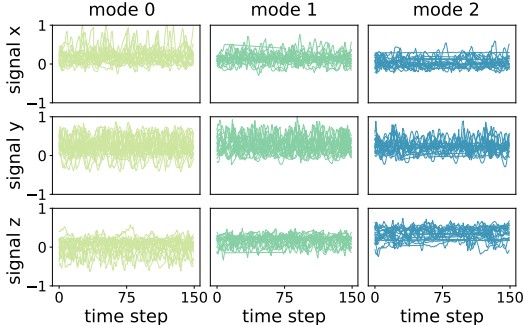

Figure 10: 3-channel accelerometer signals of three walking modes (20 random samples per mode with xyz channels). The x-axis indicates time steps, and the y-axis indicates normalized signals.

representations show compactness within each activity, separation between different activities, and partition of different walking modes, indicating independence from user ID and informativeness of activity by encoding mode information.

**Additional Analysis.** Clustering performance, computational complexity, and parameter sensitivity are analyzed in Appendix I, which show that SD-HC is (1) capable of capturing the underlying modes with k-means clustering, (2) computationally efficient w.r.t. the number of parameters, and (3) not particularly sensitive to changes of $N_m$ when it is slightly above the ground truth value, which is probably because SD-HC can preserve mode information to some extent, as long as the samples within one estimated cluster mostly belong to the same ground-truth mode.

## 6 CONCLUSIONS

In this paper, we propose a novel supervised disentanglement method, SD-HC, that deals with hidden correlations under certain attributes. We introduce mode-based CMI minimization to achieve disentanglement for these certain attributes with multi-modality and hidden correlations, and theoretically prove its sufficiency. Our results can be extended to show the general sufficiency of CMI minimization for disentanglement, demonstrating broad significance. Experiments on the toy dataset and five real-world datasets demonstrate the superiority of SD-HC along with comprehensive investigations.

## 7 ETHICS STATEMENTS

Our paper mainly focuses on scientific research of supervised disentangled representation learning and there is no potential ethical risk.

## 8 REPRODUCIBILITY STATEMENTS

We have provided the details regarding computational platforms, dataset descriptions, network architectures, hyper-parameter settings, and the training process of our method in Section 5.1 in the main paper and Appendix D, E, F, and G. Our codes are released at anonymous Github (`https://anonymous.4open.science/r/SD-HC`) as stated in the abstract. The download links of the public datasets are provided in the project homepage and pre-processing functions are included in the codes. The hyper-parameter settings are given in Appendix G.

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

## A    DATA GENERATION PROCESS UNDER ATTRIBUTE CORRELATIONS

Following (Suter et al., 2019), the data generation process under attribute correlations is formulated in Definition 3. The causal graph of Definition 3 is depicted in Figure 11.

**Definition 3.** (Disentangled Causal Process). *Consider a causal generative model $p(\boldsymbol{x}|\boldsymbol{a})$ for data $\boldsymbol{x}$ with $K$ attributes $\boldsymbol{a} = (a_1, a_2, ..., a_K)$ as the generative factors, where $\boldsymbol{a}$ could be influenced by $L$ confounders $\boldsymbol{c} = (c_1, ..., c_L)$. This causal model is called disentangled if and only if it can be described by a structural causal model (SCM) (Pearl, 2009) of the form:*

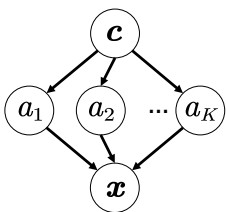

$$\boldsymbol{c} \leftarrow \boldsymbol{n}_c$$
$$a_i \leftarrow h_i(S_i^c, \boldsymbol{n}_i), S_i^c \subset \{c_1, ..., c_L\}, i = 1, ..., K \quad (7)$$
$$\boldsymbol{x} \leftarrow g(\boldsymbol{a}, \boldsymbol{n}_x)$$

with functions $g, h_i$, jointly independent noise variables $\boldsymbol{n}_c, \boldsymbol{n}_x, \boldsymbol{n}_i$, and confounder subsets $S_i^c$ for $i = 1, ..., K$. Note that $\forall i \neq j, a_i \not\rightarrow a_j$.

Figure 11: Causal graph of data generation process with attribute correlations.

## B    PROOFS

We give the complete proof of the decomposition and propositions in the main paper using knowledge of mutual information and entropy.

Note that we use formal definitions of mutual information, where separators **semicolon ";"** and **comma ","** should be distinguished from each other. Semicolon ";" separates groups of variables whose mutual information with respect to each other is being measured, while comma "," denotes the joint distribution of the listed variables.

### B.1    PROOF OF TOTAL HIDDEN CORRELATION

**Total Hidden Correlation** $I(m_1; a_2) = I(a_1; a_2) + I(m_1; a_2|a_1)$

*Proof.* Firstly, we prove $I(m_1; a_2) = I(m_1, a_1; a_2)$. Since each mode falls under one particular attribute value, the value of attribute is fully determined given the modes, i.e., $H(a_1|m_1) = 0$. Therefore, $H(a_1|m_1) = H(a_1|m_1, a_2) + I(a_1; a_2|m_1) = 0$, and followingly $I(a_1; a_2|m_1) = 0$, as both terms are non-negative. Hence $H(a_2|m_1) = H(a_2|m_1, a_1) + I(a_1; a_2|m_1) = H(a_2|m_1, a_1)$. Therefore, we have:

$$I(m_1, a_1; a_2) = H(a_2) - H(a_2|m_1, a_1)$$
$$= H(a_2) - H(a_2|m_1)$$
$$= I(m_1; a_2)$$

Secondly, we prove $I(m_1, a_1; a_2) = I(a_1; a_2) + I(m_1; a_2|a_1)$ by chain rule of mutual information:

$$I(m_1, a_1; a_2) = H(a_2) - H(a_2|m_1, a_1)$$
$$= H(a_2) - H(a_2|a_1) + H(a_2|a_1) - H(a_2|m_1, a_1)$$
$$= I(a_1; a_2) + I(m_1; a_2|a_1)$$

Finally, we reach $I(m_1; a_2) = I(m_1, a_1; a_2) = I(a_1; a_2) + I(m_1; a_2|a_1)$

### B.2    PROOF OF PROPOSITION 1

**Proposition 1.** *For representations $\boldsymbol{z}_1, \boldsymbol{z}_2$ of $m_1, a_2$, respectively, if $I(m_1; a_2|a_1) > 0$, then enforcing $I(\boldsymbol{z}_1; \boldsymbol{z}_2|a_1) = 0$ leads to at least one of $I(\boldsymbol{z}_1; m_1) < H(m_1)$ and $I(\boldsymbol{z}_2; a_2) < H(a_2)$.*

*Proof.* We prove by contradiction. Assuming $I(\boldsymbol{z}_1; m_1) = H(m_1)$ and $I(a_2; \boldsymbol{z}_2) = H(a_2)$ both stand, we have $H(m_1|\boldsymbol{z}_1) = 0$ and $H(a_2|\boldsymbol{z}_2) = 0$.

Firstly, we prove that this leads to $I(m_1; a_2; \boldsymbol{z}_1; \boldsymbol{z}_2|a_1) > 0$ with (1)(2)(3).

(1) Since $H(m_1|\boldsymbol{z}_1) = 0$ and $H(m_1|\boldsymbol{z}_1) - H(m_1|a_1, \boldsymbol{z}_1) = I(m_1; a_1|\boldsymbol{z}_1) \geq 0$ by definition of conditional mutual information, we have $0 \leq H(m_1|a_1, \boldsymbol{z}_1) \leq H(m_1|\boldsymbol{z}_1) = 0$, we have $H(m_1|a_1, \boldsymbol{z}_1) = 0$. By definition, $H(m_1|a_1, \boldsymbol{z}_1) = H(m_1|a_1, a_2, \boldsymbol{z}_1) + I(m_1; a_2|a_1, \boldsymbol{z}_1) = 0$, which gives $I(m_1; a_2|a_1, \boldsymbol{z}_1) = 0$, as both terms are non-negative. Therefore:

$$I(m_1; a_2; \boldsymbol{z}_1|a_1) = I(m_1; a_2|a_1) - I(m_1; a_2|a_1, \boldsymbol{z}_1)$$
$$= I(m_1; a_2|a_1) > 0$$

(2) Similar to (1), since $H(a_2|\boldsymbol{z}_2) = 0$ and $0 \leq H(a_2|a_1, \boldsymbol{z}_2) \leq H(a_2|\boldsymbol{z}_2) = 0$, we have $H(a_2|a_1, \boldsymbol{z}_2) = 0$. By definition, $H(a_2|a_1, \boldsymbol{z}_2) = H(a_2|m_1, a_1, \boldsymbol{z}_2) + I(m1; a2|a_1, \boldsymbol{z}_2) = 0$, which gives $I(m_1; a_2|a_1, \boldsymbol{z}_2) = 0$, as both terms are non-negative. Therefore:

$$I(m_1; a_2; \boldsymbol{z}_2|a_1) = I(m_1; a_2|a_1) - I(m_1; a_2|a_1, \boldsymbol{z}_2)$$
$$= I(m_1; a_2|a_1) > 0$$

(3) Given $H(m_1|\boldsymbol{z}_1) = 0$, we have $H(m_1|\boldsymbol{z}_1) = H(m_1|\boldsymbol{z}_1, \boldsymbol{z}_2) + I(m_1; \boldsymbol{z}_2|\boldsymbol{z}_1) = 0$ and thus $H(m_1|\boldsymbol{z}_1, \boldsymbol{z}_2) = 0$, as both terms are non-negative. Similar to (1) that yields $I(m_1; a_2; \boldsymbol{z}_1|a_1) = I(m_1; a_2|a_1)$ from $H(m_1|\boldsymbol{z}_1) = 0$, we can get $I(m_1; a_2; \boldsymbol{z}_1|a_1, \boldsymbol{z}_2) = I(m_1; a_2|a_1, \boldsymbol{z}_2)$ from $H(m_1|\boldsymbol{z}_1, \boldsymbol{z}_2) = 0$ by additionally conditioning on $\boldsymbol{z}_2$. Combined with $I(m_1; a_2; \boldsymbol{z}_2|a_1) > 0$ in (2), we have:

$$I(m_1; a_2; \boldsymbol{z}_1; \boldsymbol{z}_2|a_1) = I(m_1; a_2; \boldsymbol{z}_1|a_1) - I(m_1; a_2; \boldsymbol{z}_1|a_1, \boldsymbol{z}_2)$$
$$= I(m_1; a_2|a_1) - I(m_1; a_2|a_1, \boldsymbol{z}_2)$$
$$= I(m_1; a_2; \boldsymbol{z}_2|a_1) > 0$$

Secondly, we prove $I(m_1; a_2; \boldsymbol{z}_1; \boldsymbol{z}_2|a_1) \leq 0$ with (4)(5)(6).

(4) Given $H(m_1|a_1, \boldsymbol{z}_1) = 0$ in (1), we have $H(m_1|a_1, \boldsymbol{z}_1) = H(m_1|a_1, \boldsymbol{z}_1, \boldsymbol{z}_2) + I(m_1; \boldsymbol{z}_2|a_1, \boldsymbol{z}_1) = 0$ and followingly, $I(m_1; \boldsymbol{z}_2|a_1, \boldsymbol{z}_1) = 0$, as both terms are non-negative. Therefore:

$$I(m_1; \boldsymbol{z}_1; \boldsymbol{z}_2|a_1) = I(m_1; \boldsymbol{z}_2|a_1) - I(m_1; \boldsymbol{z}_2|a_1, \boldsymbol{z}_1)$$
$$= I(m_1; \boldsymbol{z}_2|a_1) \geq 0$$

(5) Since $I(\boldsymbol{z}_1; \boldsymbol{z}_2|a_1) = 0$, we have:

$$I(m_1; \boldsymbol{z}_1; \boldsymbol{z}_2|a_1) = I(\boldsymbol{z}_1; \boldsymbol{z}_2|a_1) - I(\boldsymbol{z}_1; \boldsymbol{z}_2|m_1, a_1)$$
$$= -I(\boldsymbol{z}_1; \boldsymbol{z}_2|m_1, a_1) \leq 0$$

(6) Combine $I(m_1; \boldsymbol{z}_1; \boldsymbol{z}_2|a_1) \geq 0$ in (4) and $I(m_1; \boldsymbol{z}_1; \boldsymbol{z}_2|a_1) \leq 0$ in (5), we have $I(m_1; \boldsymbol{z}_1; \boldsymbol{z}_2|a_1) = 0$. Given $H(m_1|a_1, \boldsymbol{z}_1) = 0$ in (1) and $H(m_1|a_1, \boldsymbol{z}_1) = H(m_1|a_1, \boldsymbol{z}_1, \boldsymbol{z}_2) + I(m_1; \boldsymbol{z}_2|a_1, \boldsymbol{z}_1)$, we have $H(m_1|a_1, \boldsymbol{z}_1, \boldsymbol{z}_2) = 0$ as both terms are non-negative. Similar to (4) that yields $I(m_1; \boldsymbol{z}_1; \boldsymbol{z}_2|a_1) = I(m_1; \boldsymbol{z}_2|a_1)$ from $H(m_1|a_1, \boldsymbol{z}_1) = 0$, we can get $I(m_1; \boldsymbol{z}_1; \boldsymbol{z}_2|a_1, a_2) = I(m_1; \boldsymbol{z}_2|a_1, a_2)$ from $H(m_1|a_1, \boldsymbol{z}_1, \boldsymbol{z}_2) = 0$ by additionally conditioning on $\boldsymbol{z}_2$. Therefore:

$$I(m_1; a_2; \boldsymbol{z}_1; \boldsymbol{z}_2|a_1) = I(m_1; \boldsymbol{z}_1; \boldsymbol{z}_2|a_1) - I(m_1; \boldsymbol{z}_1; \boldsymbol{z}_2|a_1, a_2)$$
$$= -I(m_1; \boldsymbol{z}_2|a_1, a_2) \leq 0$$

This is contradictory with $I(m_1; a_2; \boldsymbol{z}_1; \boldsymbol{z}_2|a_1) > 0$. Therefore, if $I(m_1; a_2|a_1) > 0$ and $I(\boldsymbol{z}_1; \boldsymbol{z}_2|a_1) = 0$, then at least one of $I(m_1; \boldsymbol{z}_1) < H(m_1)$ and $I(a_2; \boldsymbol{z}_2) < H(a_2)$ must hold.

### B.3 PROOF OF PROPOSITION 2

**Proposition 2.** *For representations $\boldsymbol{z}_1, \boldsymbol{z}_2$ of $m_1, a_2$, respectively, if $I(\boldsymbol{z}_1; m_1) = H(m_1)$, $I(\boldsymbol{z}_2; a_2) = H(a_2)$, and $I(\boldsymbol{z}_1; \boldsymbol{z}_2|m_1) = 0$, then $I(\boldsymbol{z}_1; \boldsymbol{z}_2) = I(m_1; a_2)$ and $I(\boldsymbol{z}_1; a_2|m_1) = 0$.*

*Proof.* First, we prove $I(m_1; a_2) \geq I(\boldsymbol{z}_1; \boldsymbol{z}_2)$ with (1)(2).

(1) Since $H(a_2|\boldsymbol{z}_2) = 0$, we have $H(a_2|\boldsymbol{z}_2) = H(a_2|\boldsymbol{z}_1, \boldsymbol{z}_2) + I(\boldsymbol{z}_1; a_2|\boldsymbol{z}_2) = 0$, and followingly $I(\boldsymbol{z}_1; a_2|\boldsymbol{z}_2) = 0$, as both terms are non-negative. Therefore, by definition of interaction information,

we have $I(\mathbf{z}_1; \mathbf{z}_2; a_2) = I(\mathbf{z}_1; a_2) - I(\mathbf{z}_1; a_2|\mathbf{z}_2) = I(\mathbf{z}_1; a_2)$. Since $I(\mathbf{z}_1; \mathbf{z}_2|m_1) = 0$, we have $I(\mathbf{z}_1; \mathbf{z}_2; a_2|m_1) = I(\mathbf{z}_1; \mathbf{z}_2|m_1) - I(\mathbf{z}_1; \mathbf{z}_2|m_1, a_2) = -I(\mathbf{z}_1; \mathbf{z}_2|m_1, a_2)$. Therefore:

$$I(\mathbf{z}_1; \mathbf{z}_2; m_1; a_2) = I(\mathbf{z}_1; \mathbf{z}_2; a_2) - I(\mathbf{z}_1; \mathbf{z}_2; a_2|m_1)$$
$$= I(\mathbf{z}_1; a_2) + I(\mathbf{z}_1; \mathbf{z}_2|m_1, a_2)$$
$$\geq I(\mathbf{z}_1; a_2)$$

(2) i. Since $H(a_2|\mathbf{z}_2) = 0$, we have $H(a_2|\mathbf{z}_2) = H(a_2|m_1, \mathbf{z}_2) + I(m_1; a_2|\mathbf{z}_2) = 0$, and followingly $I(m_1; a_2|\mathbf{z}_2) = 0$, as both terms are non-negative.

ii. Since $H(m_1|\mathbf{z}_1) = 0$, we have $H(m_1|\mathbf{z}_1) = H(m_1|\mathbf{z}_1, \mathbf{z}_2) + I(m_1; \mathbf{z}_2|\mathbf{z}_1) = 0$, and followingly $H(m_1|\mathbf{z}_1, \mathbf{z}_2) = 0$, as both terms are non-negative. Therefore, $H(m_1|\mathbf{z}_1, \mathbf{z}_2) = H(m_1|\mathbf{z}_1, \mathbf{z}_2, a_2) + I(m_1; a_2|\mathbf{z}_1, \mathbf{z}_2) = 0$, and followingly $I(m_1; a_2|\mathbf{z}_1, \mathbf{z}_2) = 0$, as both terms are non-negative.

iii. Given $I(m_1; a_2|\mathbf{z}_2) = 0$ in i. and $I(m_1; a_2|\mathbf{z}_1, \mathbf{z}_2) = 0$ in ii. as shown above, we have $I(m_1; a_2; \mathbf{z}_1|\mathbf{z}_2) = I(m_1; a_2|\mathbf{z}_2) - I(m_1; a_2|\mathbf{z}_1, \mathbf{z}_2) = 0$.

iv. Since $H(m_1|\mathbf{z}_1) = 0$, by definition of conditional mutual information, we have $H(m_1|\mathbf{z}_1) = H(m_1|\mathbf{z}_1, a_2) + I(m_1; a_2|\mathbf{z}_1) = 0$, and followingly $I(m_1; a_2|\mathbf{z}_1) = 0$, as both terms are non-negative. Thus $I(m_1; a_2; \mathbf{z}_1) = I(m_1; a_2) - I(m_1; a_2|\mathbf{z}_1) = I(m_1; a_2)$.

Given $I(m_1; a_2; \mathbf{z}_1) = I(m_1; a_2)$ in iv. and $I(m_1; a_2; \mathbf{z}_1|\mathbf{z}_2) = 0$ in iii., we have:

$$I(\mathbf{z}_1; \mathbf{z}_2; m_1; a_2) = I(m_1; a_2; \mathbf{z}_1) - I(m_1; a_2; \mathbf{z}_1|\mathbf{z}_2)$$
$$= I(m_1; a_2)$$

Given (1)(2), we have $I(m_1; a_2) = I(\mathbf{z}_1; \mathbf{z}_2; m_1; a_2) \geq I(\mathbf{z}_1; a_2)$

(3) We prove $I(\mathbf{z}_1; a_2) \geq I(m_1; a_2)$ as follows.

i. Since $H(m_1|\mathbf{z}_1) = 0$, we have $H(m_1|\mathbf{z}_1) = H(m_1|\mathbf{z}_1, a_2) + I(m_1; a_2|\mathbf{z}_1) = 0$, and followingly $I(m_1; a_2|\mathbf{z}_1) = 0$, as both terms are non-negative. Thus, by chain rule of mutual information, we have:

$$I(m_1, \mathbf{z}_1; a_2) = I(\mathbf{z}_1; a_2) + I(m_1; a_2|\mathbf{z}_1)$$
$$= I(\mathbf{z}_1; a_2)$$

ii. We also have:

$$I(m_1, \mathbf{z}_1; a_2) = I(m_1; a_2) + I(\mathbf{z}_1; a_2|m_1)$$
$$\geq I(m_1; a_2)$$

Given $I(m_1, \mathbf{z}_1; a_2) = I(\mathbf{z}_1; a_2)$ in i. and , $I(m_1, \mathbf{z}_1; a_2) \geq I(m_1; a_2)$ in ii., we have $I(\mathbf{z}_1; a_2) \geq I(m_1; a_2)$.

(4) Finally, given $I(m_1; a_2) \geq I(\mathbf{z}_1; a_2)$ with (1)(2) and $I(\mathbf{z}_1; a_2) \geq I(m_1; a_2)$ in (3), the equality must hold that $I(\mathbf{z}_1; a_2) = I(m_1; a_2)$.

Moving forward, given $I(m_1, \mathbf{z}_1; a_2) = I(\mathbf{z}_1; a_2) = I(m_1; a_2) + I(\mathbf{z}_1; a_2|m_1)$ in (3) and $I(\mathbf{z}_1; a_2) = I(m_1; a_2)$ at which we just arrived, we have $I(\mathbf{z}_1; a_2|m_1) = 0$.

### B.4 PROOF OF PROPOSITION 3

**Proposition 3.** *Under the data generation assumption of Definition 1 ($K = 2$, $k = 1$) with independent mechanisms, if $I(\mathbf{z}_1; a_2|m_1) = 0$ for representation $\mathbf{z}_1$, then $p(\mathbf{z}_1|\mathrm{do}(a_2)) = p(\mathbf{z}_1)$.*

*Proof.* We prove this by applying Rule 3 of do-calculus based on the causal graph $G$ in Figure 12, which reflects the representation learning process. The rules of do-calculus are elaborated in Appendix C.2, where $\perp\!\!\!\perp$ indicates independence between variables, for arbitrary disjoint sets of nodes $X, Z, W$, $G_{\overline{X}}$ denotes the graph obtained by deleting all arrows pointing to $X$-nodes from G, and $Z(W)$ denotes the subset of $Z$-nodes that are not ancestors of any $W$-node.

Specifically, we unfold the left-hand side of $p(\boldsymbol{z}_1|\text{do}(a_2)) = p(\boldsymbol{z}_1)$ and reach the right-hand side as:

$$p(\boldsymbol{z}_1|\text{do}(a_2)) = \sum_{m_1} p(\boldsymbol{z}_1|\text{do}(a_2), m_1)p(m_1|\text{do}(a_2)) \qquad \text{(i)}$$

$$= \sum_{m_1} p(\boldsymbol{z}_1|m_1)p(m_1) \qquad \text{(ii)}$$

$$= p(\boldsymbol{z}_1) \qquad \text{(iii)}$$

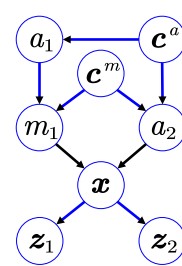

Figure 12: Causal graph of representation learning.

where we arrive at (i) by chain rule of probability, and then arrive at (ii) by using Rule 3 of do-calculus twice: **First**, given $I(\boldsymbol{z}_1; a_2|m_1) = 0$, we have $\boldsymbol{z}_1 \perp\!\!\!\perp a_2|m_1$ in $G$, as the mutual information between variables equals zero if and only if they are independent; for $G_{\overline{a_2(m_1)}} = G_{\overline{a_2}}$ (obtained by removing the edges pointing to $a_2$ from confounders $\boldsymbol{c}^a, \boldsymbol{c}^m$ in $G$), this conditional independence still holds for the following reasons (Pearl, 2009): For $\boldsymbol{z}_1$ and $a_2$, such edge removal (1) leaves the direct path $a_2 \to \boldsymbol{x} \to \boldsymbol{z}_1$ intact, not introducing any new pathway, and (2) blocks the backdoor paths $a_2 \leftarrow \boldsymbol{c}^m \to m_1 \to \boldsymbol{x} \to \boldsymbol{z}_1$ and $a_2 \leftarrow \boldsymbol{c}^a \to a_1 \to m_1 \to \boldsymbol{x} \to \boldsymbol{z}_1$, thus further reducing potential dependencies between $\boldsymbol{z}_1$ and $a_2$; now we satisfy the condition of Rule 3 and apply do-calculous as:

$$p(\boldsymbol{z}_1|\text{do}(a_2), m_1) = p(\boldsymbol{z}_1|m_1) \quad \text{Rule 3 by } \boldsymbol{z}_1 \perp\!\!\!\perp a_2|m_1 \text{ in } G_{\overline{a_2}} \text{ (representation learning)}$$

**Second**, given the *independent mechanism assumption* in Definition 1 that attributes are casually independent as in Figure 12, we satisfy the condition of Rule 3 and apply do-calculous as:

$$p(m_1|\text{do}(a_2)) = p(m_1) \qquad \text{Rule 3 by } m_1 \perp\!\!\!\perp a_2 \text{ in } G_{\overline{a_2}} \text{ (independent mechanisms)}$$

Finally, we arrive at (iii) by chain rule of probability.

*Discussions.* Our proof mainly relies on two conditions: (1) the causal independence between $m_1$ and $a_2$, which comes from the *independent mechanism assumption* (Schölkopf et al., 2012) of data, and (2) conditional independence $I(\boldsymbol{z}_1; a_2|m_1) = 0$, which is enforced upon $\boldsymbol{z}_1$ by representation learning that minimizes mode-based CMI, as proved in Proposition 2. Thereby, we conclude that *for data generated by independent mechanisms, disentangled representations can be learned by mode-based CMI minimization and supervised learning.*

### B.5 GENERALIZATION OF THEORETICAL RESULTS

Our theoretical results, including the necessary condition and the sufficient condition for disentanglement, can be generalized to multiple attributes. The extension mainly involves replacing $m_1, \boldsymbol{z}_1$ with $m_k, \boldsymbol{z}_k$, and replacing $a_2, \boldsymbol{z}_2$ with the joint $a_{-k}, \boldsymbol{z}_{-k}$, as the properties of mutual information and causal graphs remain the same for joint variables.

Specifically, the necessary condition in Figure 3 is extended to $K$ attributes in Figure 13, where the necessary condition for disentanglement under hidden correlations and potential attribute correlations is $I(\boldsymbol{z}_k; \boldsymbol{z}_{-k}|m_k) = 0$.

For the sufficient condition of disentanglement, we extend Proposition 2 to Corollary 2.1 for $K > 2$. The constraint $I(a_k; \boldsymbol{z}_k) = H(a_k)$ is added, yet this is implied in $I(\boldsymbol{z}_k; m_k) = H(m_k)$, because each mode falls under only one attribute value, and the value of the attribute is determined knowing the mode. In other words, the information contained in $a_k$ is already contained in $m_k$. In addition, the joint constraint $I(\boldsymbol{z}_{-k}; a_{-k}) = H(a_{-k})$ is broken down for each $i \neq k$, i.e., $I(\boldsymbol{z}_i; a_i) = H(a_i), i \neq k$.

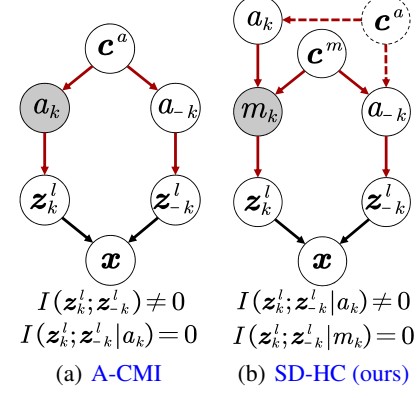

$$I(\boldsymbol{z}_k^l; \boldsymbol{z}_{-k}^l) \neq 0 \qquad I(\boldsymbol{z}_k^l; \boldsymbol{z}_{-k}^l|a_k) \neq 0$$
$$I(\boldsymbol{z}_k^l; \boldsymbol{z}_{-k}^l|a_k) = 0 \qquad I(\boldsymbol{z}_k^l; \boldsymbol{z}_{-k}^l|m_k) = 0$$

(a) A-CMI  (b) SD-HC (ours)

Figure 13: Causal graphs of the true latent representations under $K > 2$. **Red** arrows indicate the *backdoor paths* between $\boldsymbol{z}_1^l$ and $\boldsymbol{z}_2^l$. The dashed circle and arrows indicate that attribute correlations may or may not exist.

**Corollary 2.1** *For representations $z_i$ of $a_i$ $(i = 1, ..., K)$, if $I(z_i; a_i) = H(a_i)$ for $i = 1, ..., K$, $I(z_k; m_k) = H(m_k)$, and $I(z_k; z_{-k}|m_k) = 0$ for a specific $1 < k < K$, then $I(z_k; a_{-k}) = I(m_k; a_{-k})$ and $I(z_k; a_{-k}|m_k) = 0$.*

where $-k$ indicates the set of attribute indices $\{j\}_{j \neq k}$.

In addition, we extend Proposition 3 to Corollary 3.1 for $K > 2$ as follows.

**Corollary 3.1.** *Under the data generation assumption of Definition 1 with independent mechanisms, if $I(z_k; a_{-k}|m_k) = 0$ for representation $z_k$, then $p(z_k|\mathrm{do}(a_{-k})) = p(z_k)$.*

## C  CAUSALITY

### C.1  D-SEPARATION AND BACKDOOR PATHS

**Overview of Causality**   We provide a summary of notions in causal graphs relevant to the analysis in Section 3.3, namely d-separation, blocking paths, and conditional independence. More details can be found in (Pearl, 2009).

Causal graphs are directed acyclic graphs, where nodes represent random variables and directed edges represent the causal relationships between two variables. The notion of d-separation forms the link between blocking paths in the causal graph and dependencies between random variables. A *path* in causal graphs is a sequence of consecutive edges. Consider two nodes $X$ and $Y$, $X$ and $Y$ are called *d-separated* by a set of nodes $Z$ if all undirected paths from $X$ to $Y$ are *blocked* by $Z$. Meanwhile, a path between $X$ and $Y$ is considered to be *blocked* by a set of nodes $Z$ if at least one of the following holds:

(1) The path contains a chain $X \to M \to Y$ with the mediator set $M$, and a node in $M$ is in $Z$.

(2) The path contains a fork $X \leftarrow U \to Y$ with the confounder set $U$, and a node in $U$ is in $Z$.

(3) The path contains a collider $X \to C \leftarrow Y$ with the collider node $C$, and neither $C$ or its descendant is in $Z$.

Finally, if $X$ and $Y$ are d-separated by the set $Z$, $X$ and $Y$ are conditionally independent given $Z$. A *backdoor path* between $X$ and $Y$ is the non-causal path between $X$ and $Y$ that contains at least one edge pointing at $X$ or $Y$, i.e. the path that flows backward from $X$ or $Y$. Backdoor paths introduce dependence between variables, thus they need to be blocked by controlling a node on these paths as in (1) and (2).

**Causal Graph Analysis Under Hidden Correlations**   Figure 3b contains three paths between $z_1$ and $z_2$. (1) The path $z_1 \to x \leftarrow z_2$ is blocked without conditioning on any variables, as long as the collider $x$ is uncontrolled. (2) The path $z_1 \leftarrow m_1 \leftarrow c^m \to a_2 \to z_2$ is blocked if any node in the confounder set $\{m_1, c^m, a_2\}$ is controlled. Since $c^m$ is unobserved, controlling either $m_1$ or $a_2$ blocks this path. (3) The path $z_1 \leftarrow m_1 \leftarrow a_1 \leftarrow c^a \to a_2 \to z_2$ is blocked if any node in the confounder set $\{m_1, a_1, c^a, a_2\}$ is controlled. Since $c^a$ is unobserved, controlling one of $m_1$, $a_1$, and $a_2$ blocks this path. To simultaneously block all undirected paths between $z_1$ and $z_2$, we need to control either $m_1$ or $a_2$, as controlling $a_1$ does not block path (2). That is to say, $z_1$ and $z_2$ are conditionally independent given either $m_1$ or $a_2$.

### C.2  RULES OF *do*-CALCULUS

Let $X$, $Y$, $Z$, and $W$ be arbitrary disjoint sets of nodes in a causal DAG $G$. *do*-calculus consists of three inference rules that permit us to map interventional and observational distributions to each other whenever certain conditions hold in the causal diagram $G$.

We denote by $G_{\overline{X}}$ the graph obtained by deleting from G all arrows pointing to nodes in $X$. Likewise, we denote by $G_{\underline{X}}$ the graph obtained by deleting from G all arrows emerging from nodes in $X$. To represent the deletion of both incoming and outgoing arrows, we use the notation $G_{\overline{X}\underline{Z}}$. The following three rules are valid for every interventional distribution compatible with $G$ (Pearl, 2016; 1995).

Table 3: Network architectures. "Discriminator($a_{in}$)" denotes discriminator with conditional input $a_{in}$. "Conv(c$i$, k$j$, s$l$)" denotes 1D convolution layer with $i$ channels, kernel size $j$, and stride $l$. "FC($i$)" denotes fully connected layer with output dimension $i$. "BN($i$)" denotes 1D batch normalization layer with feature dimension $i$. "AvgPool($i$)" denotes 1D adaptive pooling layer with output dimension $i$. "LeakyReLU($\alpha$)" denotes LeakyReLU activations with scale $\alpha$. Output dimension $d_{out}$ is set according to each prediction task. $N_1^c$ and $N_2^c$ denote the number of values for $a_1$ and $a_2$, respectively.

| Component | Method | Dataset | Architectures |
|---|---|---|---|
| Encoder subnetwork | All | Toy | FC(16) $\rightarrow$ FC(16) |
| Encoder subnetwork | All | CMNIST | FC(128), BN(128) $\rightarrow$ FC(128), BN(128) |
| Encoder subnetwork | All | WHAR | Conv(c128, k8, s2), BN(128) $\rightarrow$ Conv(c256, k5, s2), BN(256) $\rightarrow$ Conv(c128, k3, s1), BN(128), AvgPool(1) |
| Encoder subnetwork | All | MFD | Conv(c64, k32, s6), BN(64) $\rightarrow$ Conv(c128, k8, s2), BN(128) $\rightarrow$ Conv(c128, k8, s2), BN(128), AvgPool(1) |
| Predictor | All | All | FC($d_{out}$), Softmax |
| Discriminator($m_1$) | SD-HC | All | $N_1^c \times$ [FC(512), LeakyReLu(0.2) $\rightarrow$ FC(1), Sigmoid] for each value of $a_1$ |
| Discriminator(-) | SD-HC-A | All | $N_2^c \times$ [FC(512), LeakyReLu(0.2) $\rightarrow$ FC(1), Sigmoid] for each value of $a_2$ |
| Discriminator(-) | SD-HC-ID | All | $N_1^c \times N_m \times$ [FC(512), LeakyReLu(0.2) $\rightarrow$ FC(1), Sigmoid] for each mode under each value of $a_1$ |
| Discriminator($a_1$, $m_1$) | SD-HC-SD | All | FC(512), LeakyReLu(0.2) $\rightarrow$ FC(1), Sigmoid |
| Middle layer | SD-HC-M | All | FC(128), BN(128) |

- **Rule 1**: *Insertion/deletion of observations*
$$P(y|do(x), z, w) = P(y|do(x), w), \quad \text{if } Y \perp\!\!\!\perp Z|X, W \text{ in } G_{\overline{X}}$$

- **Rule 2**: *Action/observation exchange*
$$P(y|do(x), do(z), w) = P(y|do(x), z, w), \quad \text{if } Y \perp\!\!\!\perp Z|X, W \text{ in } G_{\overline{X}\underline{Z}}$$

- **Rule 3**: *Insertion/deletion of actions*
$$P(y|do(x), do(z), w) = P(y|do(x), w), \quad \text{if } Y \perp\!\!\!\perp Z|X, W \text{ in } G_{\overline{XZ(W)}}$$

where $\perp\!\!\!\perp$ indicates independence, and for $G_{\overline{XZ(W)}}$, $Z(W)$ denotes the set of $Z$-nodes that are not ancestors of any $W$-node in $G_{\overline{X}}$.

## D NETWORK ARCHITECTURES

The detailed architectures of different components in SD-HC and its variants are summarized in Table 3. For independent control of each attribute, encoder $F$ uses individual subnetworks for each attribute with the same architectures. Predictors $C_i$, $C_i^m$ share the same architectures as well. Different architectures of discriminator $D_k$ in SD-HC, SD-HC-A, SD-HC-ID, and SD-HC-SD, and the architecture of the middle layer in SD-HC-M are described separately. SD-HC-M calculates the mode prediction loss on the output representations $z_k$ of encoder $F$ with $C_k^m$, passes $z_k$ to the middle layer, and calculates the attribute prediction loss on the output representations of the middle layer with $C_k$.

## E TRAINING PROCESS

The training process of SD-HC under $K = 2$ ($a_1$ as the attribute with multi-modality) is summarized in Algorithm 1, where optimizations w.r.t. different losses are performed alternatively. The algorithm can be generalized to multiple attributes accordingly.

## F DETAILS OF EXPERIMENTAL SETTINGS

### F.1 DATASETS

**Toy Dataset** Our 2-dimensional toy data have two binary attributes, with the primary attribute $a_1$ having 3 modes under each attribute value, i.e., $a_1 = 0, m_1 = 0, 1, 2$ and $a_1 = 1, m_1 = 3, 4, 5$.

---

**Algorithm 1** The training process of SD-HC under $K = 2$

---

1: **Input:** Training set $\mathcal{D}$ with data $\boldsymbol{x}$ and attributes labels $\boldsymbol{a} = (a_1, a_2)$, the number of modes $N_m$ under each value of $a_1$, the number of epochs $E_1$ and $E_2$, and the number of steps $S_\mathrm{d}$, $S_\mathrm{f}$, and $S_\mathrm{c}$
2: Initialize encoder $F^*$ and predictor $C_1^*$
3: **for** $epoch = 1$ **to** $E_1$ **do**
4:     **for** mini-batch $(\boldsymbol{x}, a_1)$ **in** $\mathcal{D}$ **do**
5:         Update $F^*$ and $C_1^*$ by minimizing $\mathcal{L}_{ac}$ in Equation 4
6:     **end for**
7: **end for**
8: Under each value of $a_1$, perform k-means clustering with the number of clusters $N_m$ on the output representations $\boldsymbol{z}_1$ of the trained encoder $F^*$, and get the estimated mode labels $m_1$
9: Initialize encoder $F$, predictors $C_1, C_2, C_1^m$, and discriminator $D_1$
10: **for** $epoch = 1$ **to** $E_2$ **do**
11:     **for** mini-batch $(\boldsymbol{x}, \boldsymbol{a})$ **in** $\mathcal{D}$ **do**
12:         **for** $step = 1$ **to** $S_\mathrm{c}$ **do**
13:             Update encoder $F$ and predictors $C_1$, $C_2$ and $C_1^m$ by minimizing $\mathcal{L}_c$ in Equation 5
14:         **end for**
15:         **for** $step = 1$ **to** $S_\mathrm{d}$ **do**
16:             Update discriminator $D_1$ by minimizing $\mathcal{L}_d$ in Equation 6
17:         **end for**
18:         **for** $step = 1$ **to** $S_\mathrm{f}$ **do**
19:             Update encoder $F$ by maximizing $\mathcal{L}_d$ in Equation 6
20:         **end for**
21:     **end for**
22: **end for**
23: **Output:** Encoder $F$ and predictor $C_1$

---

Table 4: Dataset descriptions.

| Dataset | UCI-HAR | RealWorld | HHAR | MFD |
|---|---|---|---|---|
| $a_1$ | activity | activity | activity | incipient fault type |
| $a_2$ | user | user | user | operating condition |
| # values of $a_1$ | 6 | 8 | 6 | 3 |
| # values of $a_2$ | 30 | 15 | 9 | 4 |
| # of groups | 5 | 5 | 3 | 4 |
| # channels | 3 | 3 | 3 | 1 |
| # samples | 11711 | 36980 | 14772 | 10916 |
| window length | 128 | 150 | 128 | 5120 |
| values of $a_1$ | walking, walking upstairs, walking downstairs, sitting, standing, laying | climbing stairs up, climbing stairs down, jumping, lying, standing, sitting, running, walking | healthy, inner-bearing damage, outer-bearing damage | |

Data are generated from the attributes as $\boldsymbol{x} = \boldsymbol{m}_1 \cdot [[0,0],[2,0],[4,0],[1,0],[3,0],[5,0]] + \boldsymbol{a}_2 \cdot [[0,0],[0,1]] + \boldsymbol{n}$, where vectors $\boldsymbol{m}_1$ and $\boldsymbol{a}_2$ represent the one-hot encoded values of $m_1$ and $a_2$, respectively, and $\boldsymbol{n} \sim \mathcal{N}(\boldsymbol{0}, \sigma^2 \boldsymbol{I})$ represents 2-dimensional independently normally distributed noise with noise level $\sigma$. For $\boldsymbol{x} = (x_1, x_2)$, the primary attribute $a_1$ and mode $m_1$ control dimension 1, i.e., $x_1$, and attribute $a_2$ controls dimension 2, i.e., $x_2$. An illustration of the generated data under different correlations and noise levels is given in Figure 14.

**CMINIST Dataset** Colored MNIST (**CMNIST**) is constructed by coloring and occluding a subset of MNIST (LeCun et al., 1998). As shown in Figure 5b, attribute $a_1$ is defined as the parity of digits, i.e., $a_1 = 0, 1$ indicates "even", "odd". Attribute $a_2$ is defined as the color of digits, i.e., $a_2 = 0, 1$ indicates "red", "blue". $a_1$ has 2 modes under each attribute value, i.e., digits 4, 2 under parity "even" and digits 3, 9 under parity "odd". Digit noises are generated as occlusion masks with occlusion ratio

Table 5: Conditional probability $p(a_2|m_1)$ on toy data for $cor_h = 0$.

| $p(a_2|m_1)$ | | $m_1$ | | | | | |
|---|---|---|---|---|---|---|---|
| | | 0 | 1 | 2 | 3 | 4 | 5 |
| $a_2$ | 0 | 0.5 | 0.5 | 0.5 | 0.5 | 0.5 | 0.5 |
| | 1 | 0.5 | 0.5 | 0.5 | 0.5 | 0.5 | 0.5 |

Table 6: Conditional probability $p(a_2|m_1)$ on toy data for $cor_h = 0.02$.

| $p(a_2|m_1)$ | | $m_1$ | | | | | |
|---|---|---|---|---|---|---|---|
| | | 0 | 1 | 2 | 3 | 4 | 5 |
| $a_2$ | 0 | 0.6 | 0.3 | 0.6 | 0.5 | 0.6 | 0.4 |
| | 1 | 0.4 | 0.7 | 0.4 | 0.5 | 0.4 | 0.6 |

Table 7: Conditional probability $p(a_2|m_1)$ on toy data for $cor_h = 0.06$.

| $p(a_2|m_1)$ | | $m_1$ | | | | | |
|---|---|---|---|---|---|---|---|
| | | 0 | 1 | 2 | 3 | 4 | 5 |
| $a_2$ | 0 | 0.7 | 0.2 | 0.6 | 0.4 | 0.7 | 0.4 |
| | 1 | 0.3 | 0.8 | 0.4 | 0.6 | 0.3 | 0.6 |

Table 8: Conditional probability $p(a_2|m_1)$ on toy data for $cor_h = 0.13$.

| $p(a_2|m_1)$ | | $m_1$ | | | | | |
|---|---|---|---|---|---|---|---|
| | | 0 | 1 | 2 | 3 | 4 | 5 |
| $a_2$ | 0 | 0.8 | 0.1 | 0.6 | 0.3 | 0.8 | 0.4 |
| | 1 | 0.2 | 0.9 | 0.4 | 0.7 | 0.2 | 0.6 |

Table 9: Conditional probability $p(a_2|m_1)$ on toy data for $cor_h = 0.28$.

| $p(a_2|m_1)$ | | $m_1$ | | | | | |
|---|---|---|---|---|---|---|---|
| | | 0 | 1 | 2 | 3 | 4 | 5 |
| $a_2$ | 0 | 0.9 | 0 | 0.6 | 0.2 | 0.9 | 0.4 |
| | 1 | 0.1 | 1 | 0.4 | 0.8 | 0.1 | 0.6 |

Table 10: Conditional probability $p(a_2|m_1)$ on toy data for $cor_h = 0.41$.

| $p(a_2|m_1)$ | | $m_1$ | | | | | |
|---|---|---|---|---|---|---|---|
| | | 0 | 1 | 2 | 3 | 4 | 5 |
| $a_2$ | 0 | 1 | 0 | 0.5 | 0.1 | 1 | 0.4 |
| | 1 | 0 | 1 | 0.5 | 0.9 | 0 | 0.6 |

Table 11: Conditional probability $p(a_2|m_1)$ on CMNIST under attribute correlations and hidden correlations.

| $p(a_2|m_1)$ | | $m_1$ | | | |
|---|---|---|---|---|---|
| | | 0 | 1 | 2 | 3 |
| $a_2$ | 0 | 0.8 | 0.05 | 0.2 | 0.95 |
| | 1 | 0.2 | 0.95 | 0.8 | 0.05 |

Table 12: Conditional probability $p(a_2|m_1)$ on CMNIST under only hidden correlations.

| $p(a_2|m_1)$ | | $m_1$ | | | |
|---|---|---|---|---|---|
| | | 0 | 1 | 2 | 3 |
| $a_2$ | 0 | $corr_p$ | $1 - corr_p$ | $corr_p$ | $1 - corr_p$ |
| | 1 | $1 - corr_p$ | $corr_p$ | $1 - corr_p$ | $corr_p$ |

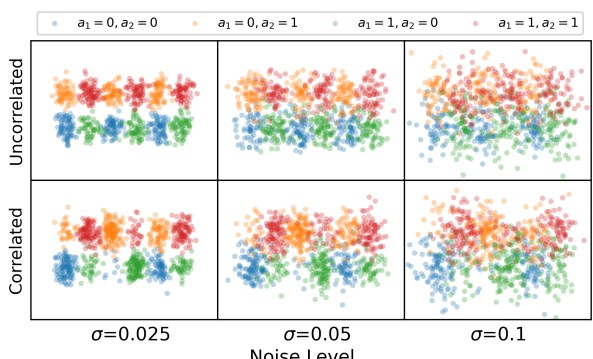

Figure 14: Generated toy data under different correlations and noise levels.

as the noise level $\sigma$ (Chai et al., 2021), and coloring noises are generated as a scalar multiplier to the RGB values of the digits.

**Time Series Datasets** We use acceleration signals from UCI-HAR, RealWorld, and HHAR datasets and vibration signals from MFD dataset. After removing invalid values and normalizing the data by channel to be within the range of [-1, 1], we pre-process the data by the sliding window strategy. For WHAR datasets with multiple sensors, we use the 3-axis acceleration data from the waist for UCI-HAR, the acceleration data from the chest for RealWorld, and the acceleration data from a Samsung smartphone for HHAR following (Ragab et al., 2023). Table 4 summarizes the statistics of the preprocessed data used in our experiments.

### F.2 EVALUATION PROTOCOL

**Toy** Since we focus on investigating the behavior of different methods under only hidden correlations $I(m_1; a_2|a_1) > 0$, data are set to be uniformly distributed under the values of $m_1$, $a_1$, and $a_2$, and attribute correlations do not exist, i.e., $I(a_1; a_2) = 0$. The hidden correlations are introduced by setting $p(a_2|m_1)$ to Table 5, 6, 7, 8, 9, 10 for hidden correlations $cor_h = 0, 0.02, 0.06, 0.13, 0.28, 0.41$, respectively.

**CMNIST** Since we focus on investigating the behavior of different methods under various correlations, data are set to be uniformly distributed under the values of $m_1$ and $a_1$. For the comparison with baselines and variants, we introduce attribute correlations and hidden correlations by setting $p(a_2|m_1)$ to Table 11. For additional analysis, we introduce hidden correlations by setting $p(a_2|m_1)$ according to Table 12, where we set $corr_p = 0.5, 0.6, 0.7, 0.8, 0.9$ for hidden correlations $cor_h = 0, 0.02, 0.08, 0.19, 0.37$, respectively.

**Time Series** Leave-one-group-out validation is performed, where each group is selected as the test group once, and the remaining groups serve as the training groups. Groups are obtained by dividing the data by the value of attribute $a_2$, where the number of values of $a_2$ is equal for different groups. The training and validation sets are obtained by splitting the data of the training groups by 0.8:0.2. All data of the test group form the test set. All methods are trained on the training set, tuned on the validation set, and tested on the test set.

### F.3 IMPLEMENTATION DETAILS

We experiment with Pytorch 1.10.0+cu113 and Python 3.8.13. Model optimization is performed using Adam (Kingma & Ba, 2015). Experiments are conducted on Linux servers with Intel(R) Core(TM) i9-12900K CPUs and NVIDIA RTX 3090 GPUs.

Table 13: Hyper-parameter search spaces and NNI settings.

| | Item | Search space / setting |
|---|---|---|
| Hyper-parameter | $w_m$ | between [0.01, 10] |
| | $S_d$ | [1, 3, 5, 7, 9] |
| | $N_m$ | [2,3,4] |
| | $l_c, l_d, l_e$ | [0.0001, 0.0003, 0.0005, 0.0007, 0.001] |
| NNI configuration | Max trial number per GPU | 1 |
| | Optimization algorithm | Tree-structured Parzen Estimator |

# G   HYPER-PARAMETERS

The general hyper-parameters are set to the following values: The number of dimensions $D$ for representations $z_i$ is set to 128. The mini-batch size is set to 64 and 128 for toy and other datasets, respectively. The number of epochs for pre-training, $E_1$, and the number of epochs for supervised DRL, $E_2$, are set to 100 and 150, respectively. The numbers of update steps $S_f$ and $S_c$ are set to 1.

Some other hyper-parameters are tuned with Neural Network Intelligence (NNI)[1]. The search spaces and NNI configurations are given in Table 13. The tuned hyper-parameters are set to the following values: The weight of mode prediction loss $w_m$ is set to 0.5, 0.2, 0.7, 0.1, 0.01, and 0.01 on toy, CMNIST, UCI-HAR, RealWorld, HHAR, and MFD for variants with mode prediction loss, respectively. The number of update steps $S_d$ is set to 2, 15, 7, 7, 1, and 1 on toy, CMNIST, UCI-HAR, RealWorld, HHAR, and MFD, respectively. The number of modes $N_m$ under each value of $a_k$ is set to 3, 2, 8, 3, 2, and 2 on toy, CMNIST, UCI-HAR, RealWorld, HHAR, and MFD, respectively. The initial learning rates of Adam $(l_c, l_d, l_e)$ are set to (0.001, 0.0007, 0.001), (0.001, 0.0003, 0.001), (0.001, 0.0007, 0.0005), (0.001, 0.001, 0.001), (0.001, 0.0001, 0.001), and (0.001, 0.001, 0.0005) on toy, CMNIST, UCI-HAR, RealWorld, HHAR, and MFD, respectively.

# H   BASELINES

We focus on comparing different independence constraints, and leave out the other components in the original baseline implementations, e.g., different architectures. For fair comparisons, all methods share the same encoder structure and train with alternative update steps, which is the same as SD-HC. The baselines are summarized below:

- **MMD** (Lin et al., 2020) minimizes the Maximum Mean Discrepancy between different distributions in the subspace of one attribute under different values of another attribute.

- **DTS** (Li et al., 2022) adversarially trains attribute predictors to make one attribute unpredictable from the representations of another.

- **IDE-VC** (Yuan et al., 2021) minimizes the unconditional MI between the representations of different attributes by adversarially training a predictor that predicts the representations of one attribute from those of another.

- **MI** (Cheng et al., 2022) and **A-CMI** (Funke et al., 2022) minimize the unconditional mutual information and the attribute-based conditional mutual information between the representations of different attributes, respectively. These two methods minimize MI by adversarially training an unconditional or conditional discriminator as the proposed method. We train two discriminators for A-CMI to minimize conditional mutual information based on both $a_1$ and $a_2$ as in (Funke et al., 2022).

- **HFS** minimizes the Hausdorff distance between two representation sets to factorize the supports of different representation subspaces, where we use Euclidean distance as the distance measure between different representations from the same subspace as in (Oublal et al., 2024).

---

[1] https://github.com/microsoft/nni

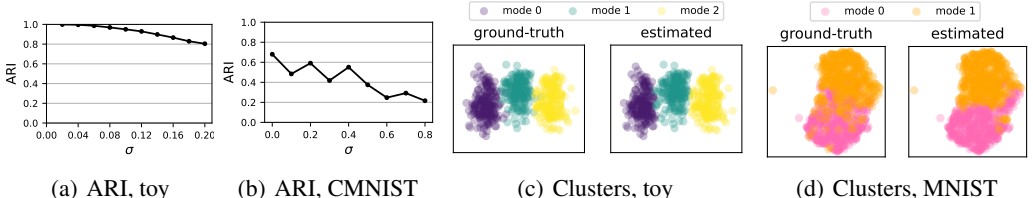

(a) ARI, toy      (b) ARI, CMNIST      (c) Clusters, toy      (d) Clusters, MNIST

Figure 15: Clustering performance. (a) and (b) shows the ARI on toy and CMNIST under varying noise levels. (c) and (d) shows the true and estimated cluster assignments under $a_1 = 0$ on the raw toy data and the CMNIST representations of BASE by t-SNE(Maaten, L. V. D. and Hinton, G., 2008).

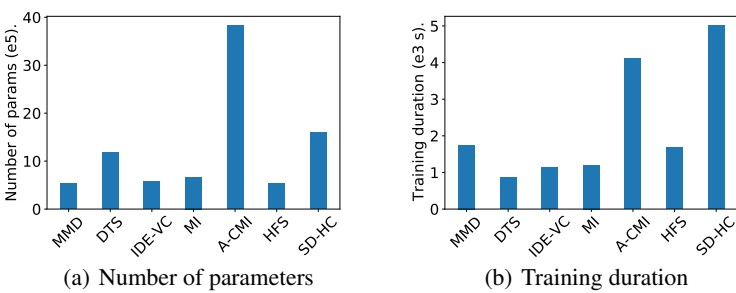

(a) Number of parameters        (b) Training duration

Figure 16: Computational complexity comparison.

# I ADDITIONAL MODEL INVESTIGATION

**Clustering Evaluation.** We use adjusted rand index (ARI) to measure clustering performance. ARI ranges between 0 and 1, with 0 indicating a random cluster assignment, and 1 indicating a perfectly matching cluster assignment with ground truth. The results are shown in Figure 15ab, which shows that: (1) As the noise level increases, ARI drops due to information loss; (2) CMNIST shows lower ARI than toy, as real data are more challenging; (3) The high ARI under moderate noises indicates effective clustering, as validated by the similarity to true cluster assignments in Figure 15cd.

The clustering algorithm is a choice of design for our method. Although k-means has been effective across our experiments, we offer practical guidance regarding the alternative clustering methods that could be considered for real applications. For high-dimensional data, Marigold (Mortensen et al., 2023) could be considered, which is an extension of k-means to high-dimensional cases. For more complex data, deep clustering (Ronen et al., 2022) could be considered, which can make representation learning and clustering mutually enhance each other by alternative training. Self-supervised learning (Zhang et al., 2019) could also be incorporated to improve the quality of representations for complex data.

**Computational Complexity** Figure 16 shows the total numbers of parameters and the training durations of a single leave-one-group-out validation process (without repetition) on UCI-HAR of SD-HC and the compared methods.

In Figure 16a, we observe that A-CMI has the most parameters, which is because A-CMI has two discriminators for minimizing conditional mutual information based on $a_1$ and $a_2$. This indicates that our method is computationally efficient w.r.t. number of parameters compared to A-CMI, which is advantageous for deployment in resource-constrained environments.

In Figure 16b, we observe that the training durations of A-CMI and SD-HC are the longest. This is because within one mini-batch, the number of samples under one mode value is much smaller than those under one attribute value, and we find that SD-HC needs more update steps to sufficiently learn the discriminator. Therefore, in real applications, the better approach is to upload the data to the server for training, and then locally download the trained network for inference.

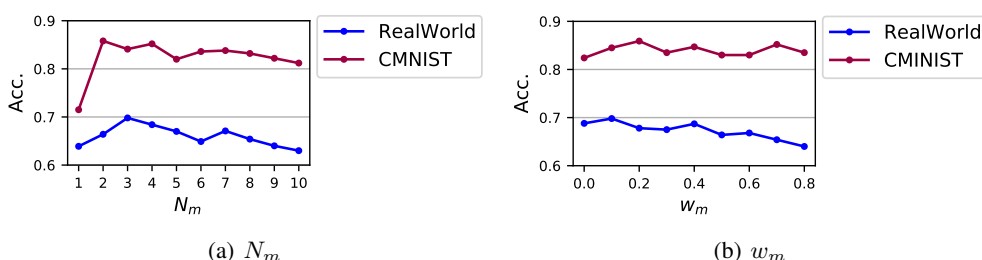

(a) $N_m$                              (b) $w_m$

Figure 17: Hyper parameter sensitivity experiments of (a) $N_m$ and (b) $w_m$ .

Table 14: Results on MINIST dataset. "*" indicates that SD-HC is statistically superior to the compared method according to the pairwise t-test at a 95% significance level. The results of the best methods are **bold**. The result of the best baseline DRL methods are underlined, over which the improvement achieved by SD-HC is calculated.

| Method | Test 1 (correlated) | | Test 2 (uncorrelated) | | Test 3 (anticorrelated) | |
|--------|---------|---------|---------|---------|---------|---------|
| | Acc. | Mac. F1 | Acc. | Mac. F1 | Acc. | Mac. F1 |
| BASE | **0.901** ±0.003 | **0.901** ±0.003 | 0.831 ±0.003 | 0.831 ±0.003 | 0.768 ±0.008 | 0.768 ±0.008 |
| MMD | 0.640 ±0.114 | 0.589 ±0.160 | 0.614 ±0.088 | 0.562 ±0.133 | 0.582 ±0.069 | 0.528 ±0.117 |
| DTS | 0.699 ±0.041 | 0.699 ±0.041 | 0.651 ±0.029 | 0.651 ±0.029 | 0.615 ±0.022 | 0.615 ±0.022 |
| IDE-VC | 0.632 ±0.031 | 0.629 ±0.031 | 0.588 ±0.025 | 0.585 ±0.028 | 0.539 ±0.022 | 0.533 ±0.027 |
| MI | 0.664 ±0.018 | 0.660 ±0.018 | 0.628 ±0.019 | 0.624 ±0.019 | 0.596 ±0.014 | 0.590 ±0.018 |
| A-CMI | 0.722 ±0.072 | 0.712 ±0.081 | 0.668 ±0.049 | 0.658 ±0.058 | 0.611 ±0.039 | 0.600 ±0.044 |
| HFS | 0.811 ±0.014 | 0.809 ±0.014 | 0.725 ±0.012 | 0.723 ±0.012 | 0.635 ±0.008 | 0.631 ±0.008 |
| SD-HC (ours) | 0.886 ±0.005 | 0.886 ±0.008 | **0.859** ±0.009 | **0.859** ±0.010 | **0.829** ±0.011 | **0.829** ±0.008 |
| **Improvement** | ↓ 1.5 | ↓ 1.5 | ↑2.8 % | ↑2.8 % | ↑6.1 % | ↑6.1 % |

**Parameter Sensitivity of $N_m$.** The sensitivity to the number of modes $N_m$ under each attribute value is shown in Figure 17a, which shows that: (1) SD-HC performs the best at the ground truth $N_m = 2$ on CMNIST, suggesting that prior knowledge about $N_m$ would be beneficial. (2) SD-HC performs badly at $N_m = 1$, where mode-based CMI degrades to attribute-based CMI, causing the loss of mode information. (3) In general, SD-HC is not particularly sensitive to changes of $N_m$ within a certain range. On CMNIST, SD-HC performs comparably under $N_m = 2, 3, 4$, suggesting that SD-HC is robust to the changes of $N_m$ when it is slightly larger than the ground truth ($N_m = 2$). Probably because as long as the samples within one estimated cluster belong to the same ground-truth mode, SD-HC can preserve mode information to some extent.

In practice, hyper-parameter tuning may come with high computational costs for large-scale datasets. Alternatively, we offer practical guidance to reduce the computational costs by estimating the number of modes $N_m$ in a data-driven manner. This requires expert knowledge to choose the suitable method: For well-separated clusters, Elbow Method (Marutho et al., 2018) would be suitable for estimating $N_m$ with k-means clustering; For complex and overlapping clusters, Bayesian Information Criterion (Watanabe, 2013) would be suitable for estimating $N_m$ with Gaussian Mixture Models for clustering; In addition, during our pre-training stage, the number of modes can be estimated by split and merge operations with deep clustering methods (Ronen et al., 2022).

**Parameter Sensitivity of $w_m$.** The sensitivity to the weight parameter of mode prediction loss, $w_m$, is shown in Figure 17b, which shows that: In general, SD-HC performs better at a small value of $w_m$. Theoretically, adding mode prediction loss benefits disentanglement. However, enforcing mode prediction with estimated mode labels will potentially introduce errors, as the estimated mode labels do not match the ground-truth mode labels.

## J    FULL RESULTS ON CMNIST DATASET

The full comparison with baselines on CMNIST dataset is presented in Table 14, from which we observe that the advantage of SD-HC increases as correlation shift increases from test 1 to test 3.

