# OpenReview forum: "Supervised Disentanglement Under Hidden Correlations"
_ICLR.cc/2025/Conference — Submitted to ICLR 2025_

### Official Review · Reviewer_HxXe · 2024-11-02

**Soundness:** 3
**Presentation:** 3
**Contribution:** 2
**Rating:** 6
**Confidence:** 3

**Summary:**

This paper addresses disentangled representation learning for data with correlated, multimodal attributes and proposes Supervised Disentanglement under Hidden Correlations (SD-HC), which infers hidden modes within attributes to achieve disentanglement while preserving essential information by minimizing mode-based conditional mutual information (CMI). The proposed approach is validated through theoretical analysis and experiments on illustrative toy data, image data (colored MNIST), and time series datasets (UCI-HAR, RealWorld, HHAR, and MFD), showing improved performance over existing methods.

**Strengths:**

(Disclosure: I've reviewed this paper before, and I used a PDF comparison tool to check the differences.)

- This paper studies a challenging problem in disentangled representation learning.
- The motivating example in Figure 1 is easy to follow and highlights the need for this method. It also links to the experiments on the human activity recognition task in Section 5.
- The symbols and technical terms are properly defined. The definitions of disentangled data generation process and disentangled representations are clearly stated in Definitions 1 and 2.
- The paper provided sufficient empirical evidence on both image and time series data.

**Weaknesses:**

- The connections between the theory and learning are still unclear to me. A better way to characterize the proposed method is to say something like "Under x assumptions, given x supervision, when x loss is optimized or x condition is satisfied, the learned representations are disentangled in the sense of definition x".
- The propositions and corollaries can be written in a more self-contained manner, even if the symbols have been defined in the text above.

**Questions:**

- (minor) Why renaming? The current name SD-HC sounds more like a problem setting, not a specific method.

---

> ### Author Response · Authors · 2024-11-21
>
> Dear Reviewer HxXe:
>
> What an honor to engage in discussion with you again! We deeply appreciate your positive acknowledgment of our problem setting, motivation, technical quality and empirical evidence. As you have checked yourself, we take reviewers' suggestions into serious consideration and put in great effort in improving our work. We highly value your comments, and are dedicated to addressing your concerns. Accordingly, we have uploaded the latest revised version, with **blue** markings indicating new or modified content.
>
>
> ### W1: Connections between theory and learning
>
>
> Thank you for your detailed suggestions. We agree that a structured description would be more clear. As suggested, we have revised **"Theoretical Contributions", Section 3.3** to explicitly outline the connection between theory and learning. For your convenience, the revised contents are provided as follows:
>
> * Formally, under the independent mechanism assumption in Definition 1, given the label supervision of all attributes and modes, when the supervised losses on all attributes and modes are optimized, and the CMI of $z_k$ ($I(z_k;z_{-k}|m_k)$ for multi-modal or $I(z_k;z_{-k}|a_k)$ for uni-modal cases) is minimized, the learned $z_k$ is disentangled in the sense of Definition 2, as elaborated in **Appendix B.5**.
>
> In addition, we have moved the connection between *Informativeness* and supervised losses to the second paragraph of **"The Sufficient Condition for Disentanglement", Section 3.3** for clarity.
>
> * *Informativeness* requires $I(z_1; a_1) = H(a_1)$ and $I(z_1; m_1) = H(m_1)$, which can be achieved by cross-entropy minimization [1].
>
> *Reference*:
>
> [1] A unifying mutual information view of metric learning: Cross-entropy vs. pairwise losses, ECCV, 2020.
>
>
> ### W2: Self-contained theoretical results
>
> We appreciate your invaluable suggestion and agree that propositions should be more self-contained. In the revised paper, we have revised the statements to ensure that each proposition and corollary is more clearly articulated, with important context and symbols restated within the propositions themselves, rather than relying on previous definitions.
>
> For your convenience, we provide the revised version as follows:
>
> * Proposition 1: For representations $z_1, z_2$ of $m_1, a_2$, respectively, if $I(m_1; a_2|a_1)>0$, then enforcing $I(z_1; z_2|a_1)=0$ leads to at least one of $I(z_1;m_1) < H(m_1)$ and $I(z_2;a_2)<H(a_2)$.
>
> * Proposition 2: For representations $z_1, z_2$ of $m_1, a_2$, respectively, if $I(z_1;m_1) = H(m_1)$, $I(z_2;a_2)=H(a_2)$, and $I(z_1;z_2|m_1)=0$, then $I(z_1;a_2)=I(m_1;a_2)$ and $I(z_1;a_2|m_1)=0$.
>
> * Proposition 3: Under the data generation assumption of Definition 1 ($K=2$, $k=1$) with independent mechanisms, if $I(z_1;a_2|m_1)=0$ for representation $z_1$, then $p(z_1| \mathrm{do} (a_{2}))=p(z_1)$.
>
>
> ### Q1: (minor) Method Name
>
>
> We gladly elaborate on the reasons for renaming as follows.
>
> * We agree with you that SD-HC also describes our problem setting. Since we are the first to raise and solve this problem, we consider it appropriate to highlight this problem in our method name.
> * Our method offers a coherent framework that involves clustering and disentangling, which can be flexibly tailored to disentangle different numbers of attributes with different forms of independence constraints according to the downstream tasks. We adopt the name SD-HC to better incorporate the overall framework and the general sense of our method.
> * In contrast, the previous name M-CMI only reflects the core loss function (mode-based conditional mutual information) of our method, yet it fails to incorporate the mode label estimation part of our method and the flexibility of our method.
>
> Thanks again for your invaluable comments. We hope our responses address your concerns and earn a more favorable perspective on our submission. Should there be further questions, we'd love to answer them!

---

> > ### Comment · Reviewer_HxXe · 2024-11-25
> >
> > I've read the response and the revised draft (except the proofs) and skimmed other reviews and responses. I appreciate your efforts to clarify the theoretical implications. I maintain my positive rating.

---

> > > ### Author Response · Authors · 2024-11-26
> > >
> > > Thank you for your thoughtful review and positive feedback! We highly value your comments and find them very insightful and constructive for linking our theoretical results with implementations and learning, which is critical for the proposed method.
> > >
> > > We express our deepest gratitude for your time, effort, and support. It is a privilege to engage in such constructive discussions with you. Meanwhile, if any aspects of our work could be further refined, we would greatly appreciate your feedback and are eager to improve our work further.
> > >
> > > Best regards

---

> > > > ### Author Response · Authors · 2024-12-01
> > > > **A supplementary note for Reviewer HxXe**
> > > >
> > > > We sincerely thank you again for your positive feedback! As a supplementary note, we'd like to provide some additional results, which relate to **the connections between our theory and learning in a more complex scenario** and might be of interest to you. Specifically, we experiment on a new toy dataset as described below, which has **multiple attributes with more than one attribute exhibiting multi-modality**, and we have gained some interesting findings closely related to our theory.
> > > >
> > > > ### Data construction
> > > >
> > > > The new toy data are 4-dimensional with **4 attributes** ($a_i, 1 \le i \le 4$), where 2 attributes ($a_1, a_2$) exhibit multi-modality. We do not introduce multi-modality for all attributes, so that the impact on the attributes **without multi-modality** can still be observed.
> > > >
> > > > The new toy dataset is an **extension of the original toy dataset** in our paper. Similar to the original toy data, each data axis is controlled by one attribute, i.e., $x_i$ is affected by $a_i$ and noises, and unaffected by other attributes $a_{-i}$. Here $a_1$ and $a_2$ with multi-modality are constructed the same as the $a_1$ in the original toy data, with 3 modes under each attribute value; $a_3$ and $a_4$ are constructed the same as the $a_2$ in the original toy data. The mappings from attribute/mode labels to the corresponding data axis are also the same.
> > > >
> > > > ### Experiment settings
> > > >
> > > > We use the same settings as the original toy data, where we train on correlated data, and evaluate on three test sets, namely test 1 with the same correlations, test 2 without correlations, and test 3 with anticorrelations. **Complex attribute and hidden correlations** are introduced on the train data, e.g., $I(a_2;a_4) = 0.07, I(a_3; a_4)=0.13, I(m_1;a_2|a_1)=0.14, I(m_1;a_4|a_1)=0.36, I(m_2;a_3|a_2)=0.28$, where mutual information $I(\cdot;\cdot)$ measures the correlations between variables.
> > > >
> > > > The task is to learn disentangled representations for each attribute. We compare SD-HC with BASE (with supervised losses only), A-CMI, and SD-HC-T (with ground-truth mode labels). For SD-HC(-T), we use mode-based conditional mutual information (CMI) minimization for $a_1$ and $a_2$ with multi-modality, and attribute-based CMI minimization for $a_3$ and $a_4$. For A-CMI, attribute-based CMI minimization is used for all attributes.
> > > >
> > > > ### Results
> > > >
> > > > |BASE|$a_1$|$a_2$|$a_3$|$a_4$|
> > > > |-|-|-|-|-|
> > > > |test 1|0.965|0.871|0.999|0.999|
> > > > |test 2|0.757|0.697|0.997|0.998|
> > > > |test 3|0.539|0.585|0.997|0.994|
> > > >
> > > > |A-CMI|$a_1$|$a_2$|$a_3$|$a_4$|
> > > > |-|-|-|-|-|
> > > > |test 1|0.510|0.593|0.998|0.525|
> > > > |test 2|0.507|0.537|0.997|0.507|
> > > > |test 3|0.507|0.518|0.996|0.476|
> > > >
> > > > |SD-HC|$a_1$|$a_2$|$a_3$|$a_4$|
> > > > |-|-|-|-|-|
> > > > |test 1|0.994|0.797|0.998|0.999|
> > > > |test 2|0.870|0.740|0.998|1.000|
> > > > |test 3|0.752|0.741|0.997|0.999|
> > > >
> > > > |SD-HC-T|$a_1$|$a_2$|$a_3$|$a_4$|
> > > > |-|-|-|-|-|
> > > > |test 1|0.999|0.997|0.997|1.000|
> > > > |test 2|0.966|0.982|0.996|1.000|
> > > > |test 3|0.928|0.964|0.998|0.999|
> > > >
> > > > The attribute prediction accuracy is reported above, which shows that:
> > > > * **A-CMI** performs badly on $a_4$, even though $a_3, a_4$ do not exhibit multi-modality and are easy to predict for the BASE method. This can be explained by our Proposition 1, which suggests that, attribute-based CMI minimization on $a_1$ loses its mode information, thus might harm the predictive ability for $a_1$ and other attributes that share hidden correlations with $m_1$, e.g., $a_4$ with $I(m_1;a_4|a_1)>0$.
> > > > * **SD-HC-T** outperforms SD-HC by a larger margin compared to the original toy data, probably because:
> > > > (1) The increased complexity of data hinders clustering for estimating $m_i$, introducing errors via the corresponding mode-based CMI and mode prediction losses based on $m_i$;
> > > > (2) The disentanglement of one representation also relies on the *Informativeness* of other representations, as shown in Proposition 2. Therefore, the disentanglement for $a_1$ might also be affected by mode label estimation errors on the other attribute $a_2$, which causes the loss of mode information and hurts the *Informativeness* of $z_2$, and vice versa.
> > > > * **SD-HC** generally shows **superiority** compared to BASE and A-CMI. For attributes $a_1,a_2$ with multi-modality, SD-HC estimates mode labels and minimizes mode-based CMI, which can preserve their mode information to some degree. For simple attributes $a_3, a_4$, SD-HC considers their hidden correlations with the modes of other attributes, which can preserve their predictive abilities.
> > > >
> > > > We believe these analyses link our theory with practical learning more closely in complex scenarios. Rest assured that these additional results will be incorporated into our paper. As the discussion phase comes to an end, we would like to sincerely thank you for your detailed and constructive comments. We kindly ask for your continued support and truly appreciate your time and thoughtful consideration.

---

### Official Review · Reviewer_wrRV · 2024-11-04

**Soundness:** 2
**Presentation:** 2
**Contribution:** 2
**Rating:** 3
**Confidence:** 3

**Summary:**

The paper introduces a method called Supervised Disentanglement under Hidden Correlations (SD-HC) to improve disentangled representation learning (DRL). Traditional DRL methods struggle with hidden correlations in multi-modal data. SD-HC addresses this by discovering data modes under certain attributes and minimizing mode-based conditional mutual information. This approach ensures disentanglement while preserving mode and attribute information. The paper provides theoretical proofs and empirical evidence showing SD-HC outperforms existing methods on various datasets.

**Strengths:**

1. The studied problem is open and important -- latent confounding is prevalent and proper efforts are deserved to this problem.
2. The experiments are extensive and the proposal looks promising.

**Weaknesses:**

1. The paper may benefit from a clear introduction of the task, especially the instantiation with real-world examples. Although I appreciate several examples present in the paper, a coherent running example would aid the readability much more if it includes the objective of the task, the problem of the hidden correlation, and the instantiation of the proposal in this example.
2. The graphical model Figure 2 is not well-defined for me. $z$ variables are functions of the observed data $x$. How would they appear in the true data-generating process? To me, this is not even a causal graph -- if I intervene on attribute $a_{i}$ in the true data-generating process, this influences will not be blocked by estimated variables $z$'s.
3. The main arguments are elusive to me. Especially, what does Proposition 2 imply? Its connection to the disentanglement is unclear. To me, it states that if you can find the modality variable $m_{1}$ and condition on it, the associated estimated variable $z_{1}$ will be d-separated from other attributes $a_{-1}$. First, there is no guarantee to discover the true modality variable $m_{1}$ (the proposal is somehow heuristic); second, how would this lead to the disentanglement conclusion?

**Questions:**

Please refer to the weaknesses section.

---

> ### Author Response · Authors · 2024-11-21
> **Response to Reviewer wrRV (Part 1 of 3)**
>
> Dear Reviewer wrRV:
>
> We sincerely thank you for the favorable recognition of the importance of the studied problem and the soundness of our experiments. We appreciate your detailed and perceptive feedback, and address your concerns with thorough and careful considerations as follows. Accordingly, we have uploaded the latest revised version, with **blue** markings indicating new or modified content.
>
>
> ### **W1: Coherent example**
>
> We agree that a coherent running example would greatly aid the readability. We'd like to point out that coherent examples in the context of wearable human activity recognition (WHAR) have been included throughout our paper, explaining our **problem settings** in the **3rd paragraph, Introduction**, and the **instantiation** w.r.t. the specific attribute settings in the **"Intuitive Example", Section 3.1**.
>
> In light of your suggestion, we add examples w.r.t. the **objective** of the task at the end of **"Intuitive Example", Section 3.1**, and further explanations on the **instantiations** w.r.t. the settings on real-world datasets in **"Datasets", Section 5.1**. For your convenience, we provide the content as follows:
>
> * **The **objective** of the task**: For activity recognition, the goal is to learn disentangled activity representations that fully capture the activity and its modes, while remaining unaffected by personalized user patterns.
>
> * **Instantiations on real-world WHAR datasets**: WHAR identifies activities with variations under each activity. Accordingly, $a_1$ represents activity, $m_1$ represents unknown activity modes, and $a_2$ represents user ID, which is often correlated with activity due to personal behavior patterns.
>
> ### **W2: Causal graph of representation learning**
>
> Thank you for pointing this out! We agree with you that *the extracted representations should not appear in the true data-generating process*. We realize that our presentation may have led to some misunderstanding, as **the $z$ in Figure 3 does not indicate the representations extracted from data**. Allow us to clarify this point:
>
> * **The meaning of $z$ in Figure 3**: This $z$ indicates **the true latent representations of attributes or modes in the underlying data generation process**. The goal of disentangled representation learning is to capture the information contained in such latent representations, which is why we call them "ideally disentangled" at first (although this might be misleading). Similar data generation graphs that involve an edge from such $z$ to $x$ include Figure 1 in [1] and Figure 1 in [2].
>
> * **Intuitive example**: For human activity data, we consider the edges in $a_1 \to m_1 \to z_1 \to x$, where $a_1$ represents the activity label (e.g., "walking"), $m_1$ represents the mode label (e.g., walking mode "stroll"), and $z_1$ captures the specific body movements that characterize the activity (e.g., the pace, stride, posture that identifies "stroll"). Along with the latent representations of other attributes, $z_1$ eventually generates the human activity data.
>
> We feel the need to highlight the difference between the true latent representations and the extracted representations, although this might be implicit in relevant works [1,2]. Revisions have been made in **"The Necessary Condition for Disentanglement", Section 3.3**, including using distinct notations for the true latent representations ($z^l$) and the extracted representations ($z$), and explicitly explaining this in plain text.
>
>
> *Reference*:
>
> [1] Auto-encoding variational bayes, arXiv preprint, 2013.
>
> [2] Hierarchical generative modeling for controllable speech synthesis, ICLR, 2019.

---

> ### Author Response · Authors · 2024-11-21
> **Response to Reviewer wrRV (Part 2 of 3)**
>
> ### **W3: From Proposition 2 to disentanglement**
>
> Thanks for this insightful comment. We'd like to address your concern from two aspects in the responses to W3.1 and W3.2.
>
> ### **W3.1: Discovery of modes**
> You are right that we cannot guarantee that the true mode labels will be discovered exactly. However, we have analyzed **clustering performance and the effect of poor clustering** with the following experiments:
>
> * Various factors might impact clustering performance. We have evaluated **clustering performance** and analyzed the **impact of noise level** in **"Clustering Evaluation", Appendix I**. We find that although clustering performance **drops as the noise level increases**, k-means has been generally **effective** across our experiments.
>
> * We have further analyzed the impact on **attribute prediction performance** in **"The Impact of Noise Level and Train Hidden Correlation", Section 5.4**, with the results in **Figure 8ab**. We find that as clustering performance drops with increasing noise levels: (1) the gap between SD-HC (based on estimated mode labels) and **SD-HC-T (based on ground-truth mode labels)** slightly increases then stabilizes; and (2) SD-HC remains superior in general. This indicates that although SD-HC's performance is **slightly affected by poor clustering on noisy data**, it still **surpasses the baselines** due to partial preservation of mode information.
>
> In addition, the clustering method is a choice of design for our method, and **clustering performance can be improved by choosing appropriate clustering algorithms** [3,4,5] for specific applications. In light of your comment, we added discussions in **"Clustering Evaluation", Appendix I** for practical guidance and mentioned this in **"Mode Label Estimation", Section 4**. Please refer to the revised paper for details.
>
> *Reference*:
>
> [3] Marigold: Efficient k-means clustering in high dimensions, VLDB, 2023.
>
> [4] Deepdpm: Deep clustering with an unknown number of clusters, CVPR 2022.
>
> [5] Self-supervised convolutional subspace clustering network, CVPR 2019.

---

> ### Author Response · Authors · 2024-11-21
> **Response to Reviewer wrRV (Part 3 of 3)**
>
> ### **W3.2: Theoretical link between Proposition 2 and disentanglement**
> In light of your comment, we provide more rigorous proof from Proposition 2 to disentanglement using do-calculous. But first, we explain why we introduce Proposition 2, and then we discuss the implications of Proposition 2 and offer a deeper understanding of disentanglement defined in Definition 2.
>
> * **Why introduce Proposition 2 from the perspective of mutual information**: As agreed upon in **W2** and supported in Figure 2 in [6], a causal graph with the extracted representations $z$ should contain the path $a \to x \to z$, starting from attributes $a$. Through representation learning, we **cannot impose causal relationships on the extracted $z$**, as $z$ is essentially learned from $x$ and inherits $a$ as its indirect causes. However, we **can control what information $z$ encodes from $x$** by designing learning objectives. Thus, we adopt mutual information to evaluate the information $z_1$ encodes about the other attribute $a_2$ in Proposition 2.
>
> * **The implications of Proposition 2**: We have **added a conclusion in Proposition 2** and provided the proof in **Appendix B.3**, which is in line with your observation *"if you can find the modality variable and condition on it, the associated estimated variable will be d-separated from other attributes"*. The added $I(z_1;a_2|m_1)=0$ implies that **$z_1$ does not contain any more information about irrelevant attributes given $z_1$'s mode**. This reflects the desired disentanglement property: $z_k$ should only encode $a_k$ and $m_k$, and changes in other attributes $a_{-k}$ should not affect $z_k$ if $m_k$ remains unchanged, which is formalized in **Definition 2** using do-operations of external intervention. Kindly note that Definition 2 is optimized by omitting data $x$ and attribute values $\alpha_{-i}$ to better focus on the effects of intervention.
>
> * **Proving disentanglement**: The definition of disentanglement in Definition 2 is a requirement on the **post-intervention distribution** of the extracted representations, which cannot be directly estimated due to unobserved confounders [6,7]. For rigorous proof from Proposition 2 to disentanglement, we use do-calculous to translate the post-intervention distribution with do-operators into expressions involving only observed data. The proof mainly relies on the **independent mechanism assumption** of data, which assumes causal independence among attributes, and **the conditional independence $I(z_1;a_2|m_1)=0$**, which can be enforced by minimizing mode-based CMI as shown in Proposition 2.
>
> Our proof is given in **Appendix B.4**, and relevant theories are provided in **Appendix C.2**. We would like to express our sincere gratitude once again for your high-level academic insights, which have been greatly beneficial in improving the rigor of our theory!
>
> *Reference*:
>
> [6] Robustly disentangled causal mechanisms: Validating deep representations for interventional robustness, ICML, 2019.
>
> [7] Desiderata for representation learning: A causal perspective, arxiv, 2021.
>
> Your comments are greatly appreciated. During the last few days, we've been dedicated to addressing your concerns with thorough and careful consideration. We hope the clarifications and supplemented proof can address your concerns and earn a more favorable perspective on our submission. If there are any remaining issues, we'd love to address them further!

---

### Official Review · Reviewer_dyTi · 2024-11-04

**Soundness:** 2
**Presentation:** 2
**Contribution:** 2
**Rating:** 5
**Confidence:** 3

**Summary:**

This paper addresses the problem of disentangled representation learning in the presence of hidden correlations, which are common in real-world data. The authors consider a multi-modal data distribution where certain attributes may induce complex modes correlated with other attributes. To tackle this issue, they first investigate the necessary and sufficient conditions for supervised disentanglement under the proposed data-generating process. Subsequently, they introduce the SD-HC method for learning disentangled representations with hidden correlations. The proposed method is validated on both synthetic data and five real-world datasets.

**Strengths:**

1. The paper studies an important problem in disentangled representation learning by considering hidden correlations, which are prevalent in real-world data.
2. The experimental evaluation is thorough, including both synthetic and real-world datasets.

**Weaknesses:**

1. The major issue lies in the lack of detailed theoretical explanation:
   - The causal graph presented in Figure 3 is confusing. Since $z = f(x)$ is a function of $x$, there should be a directed edge from $x$ to $z$. However, the graph shows $z$ as a parent of $x$, which is misleading.
   - The claim that "mode-based conditional mutual information (CMI) minimization is the necessary and sufficient condition for supervised disentanglement" lacks a detailed formal list of necessary assumptions. This claim heavily depends on the data-generating process, and the specific assumptions should be clearly highlighted.
   - The link between Proposition 2 and the independence requirement in Definition 2 is unclear. Proposition 2 appears to be a mutual information result that is not directly related to causality. Although the authors attempt to bridge this gap in Lines 242 to 244, a more rigorous analysis with detailed assumptions and proofs is needed.
2. The mode label estimation step is confusing. In this step, mode labels are estimated by clustering on the corresponding attribute. Consequently, there would be no confounders that affect $m_k$ and $a_{-k}$ while not affecting $a_k$, which violates the assumptions depicted in Figures 2 and 3.
3. The paper lacks detailed explanations regarding the choice of attributes and the modes in the real-world datasets. Providing more context would help in understanding the experimental results better.
4. There are typos in Lines 332 and 336. "Figure 7a" and "Figure 7b" should be replaced with "Figure 5a" and "Figure 5b," respectively.

**Questions:**

See the weakness part.

---

> ### Author Response · Authors · 2024-11-21
> **Response to Reviewer dyTi (Part 1 of 3)**
>
> Dear Reviewer dyTi:
>
> Thank you for your careful review and constructive feedback. We deeply appreciate your positive acknowledgment of our work's novel problem setting and comprehensive experiments. We are especially grateful for your detailed comments and invaluable suggestions. Accordingly, we have uploaded the latest revised version, with **blue** markings indicating new or modified content.
>
> ### W1.1: Theoretical explanation of Figure 3
>
> Thank you for pointing this out! We fully agree with your professional opinion that *there should be a directed edge from $x$ to the extracted representations $z$, which is a function of $x$*. We realize that our presentation may have led to some misunderstanding, as **the $z$ in Figure 3 does not indicate the representations extracted from data**. Allow us to clarify this point:
>
> * **The meaning of $z$ in Figure 3**: This $z$ indicates **the true latent representations of attributes or modes in the underlying data generation process**. The goal of disentangled representation learning is to capture the information contained in such latent representations, which is why we call them "ideally disentangled" at first (although this might be misleading). Similar data generation graphs that involve an edge from such $z$ to $x$ include Figure 1 in [1] and Figure 1 in [2].
>
> * **Intuitive example**: For human activity data, we consider the edges in $a_1 \to m_1 \to z_1 \to x$, where $a_1$ represents the activity label (e.g., "walking"), $m_1$ represents the mode label (e.g., walking mode "stroll"), and $z_1$ captures the specific body movements that characterize the activity (e.g., the pace, stride, posture that identifies "stroll"). Along with the latent representations of other attributes, $z_1$ eventually generates the human activity data.
>
> We feel the need to highlight the difference between the true latent representations and the extracted representations, although this might be implicit in relevant works [1,2]. Revisions have been made in **"The Necessary Condition for Disentanglement", Section 3.3**, including using distinct notations for the true latent representations ($z^l$) and the extracted representations ($z$), and explicitly explaining this in plain text.
>
> *Reference:*
>
> [1] Auto-encoding variational bayes, arXiv preprint, 2013.
>
> [2] Hierarchical generative modeling for controllable speech synthesis, ICLR, 2019.
>
> ### W1.2: List of assumptions
>
> Thank you for your insightful comment. The claim *"mode-based conditional mutual information (CMI) minimization is the necessary and sufficient condition for supervised disentanglement"* **does depend on the data generation assumptions in Definition 1**, which mainly relies on the Independent mechanism assumption as follows:
>
> * **Independent mechanisms [3]**: Attributes are casually independent, i.e., each attribute arises on its own, allowing changes or interventions on one attribute without affecting others. Still, confounding may exist.
>
> However, the data generation process **does not** strictly depend on any assumptions in the following aspects:
>
> * **The number of modes**: We use variable $m_k$ to model the possible multi-modal data distribution under a certain attribute $a_k$. This can be generalized to the uni-modal case, where the number of modes $N_m$ under each value of $a_k$ reduces to 1, and mode-based CMI degrades to attribute-based CMI.
>
> * **Correlation strengths**: We use confounders to model attribute correlations and hidden correlations. Confounders may induce correlations of any strength, including the strongest correlation (indicating one-to-one correspondence) and no correlation (indicating independence). When hidden correlations are absent, our method performs similarly to A-CMI; as hidden correlations strengthen, our method becomes more advantageous, as shown in **Figure 8cd** and proved in **Proposition 1**.
>
> We consider the above two points as **advantages** of our method: While mode-based CMI minimization excels under multiple modes and strong hidden correlations, **the proof generalizes to uni-modal and uncorrelated cases**, ensuring effectiveness across complex and simple scenarios without strict assumptions. We have made revisions to clarify this in **"Data Generation Process", Section 3.1**, the first paragraph in **Section 3.3**, and **"Generalization to Multiple Attributes and Simple Cases", Section 3.3**.
>
> *Reference:*
>
> [3] On Causal and Anticausal Learning, ICML, 2012.

---

> ### Author Response · Authors · 2024-11-21
> **Response to Reviewer dyTi (Part 2 of 3)**
>
> ### W1.3: Link between Proposition 2 and Definition 2
>
> We sincerely appreciate your scientific rigor, and fully concur that *Proposition 2 is a mutual information result that is not directly related to causality*. As suggested, we provide more **rigorous proof**. In addition, we'd like to provide more insights regarding Proposition 2 and disentanglement.
>
> * **Why introduce Proposition 2 from the perspective of mutual information**: As agreed upon in **W1.1** and supported in Figure 2 in [4], a causal graph with the extracted representations $z$ should contain the path $a \to x \to z$, starting from attributes $a$. Through representation learning, we **cannot impose causal relationships on the extracted $z$**, as $z$ is essentially learned from $x$ and inherits $a$ as its indirect causes. However, we **can control what information $z$ encodes from $x$** by designing learning objectives. Thus, we adopt mutual information to evaluate the information $z_1$ encodes about the other attribute $a_2$ in Proposition 2.
>
> * **A deeper understanding of the definition of disentanglement**: We adopt the definition of disentanglement in [4,5], which states that *if the probability distribution of $z_i$ stays unaffected by external interventions on other attributes $a_{-i}$*, then $z_i$ fits the independence requirement of disentanglement.
> Note that this is a requirement on the **post-intervention distribution** of $z_i$, rather than the causal relationship. Interventions with do-operations help us determine how variables are affected by the changes of other variables, and such effects might vary for different observed data with the same causal graph structure. Kindly note that **Definition 2** is optimized by omitting data $x$ and attribute values $\alpha_{-i}$ to better focus on the effects of intervention.
>
> * **Proving disentanglement**: For rigorous proof, we use do-calculous to translate the post-intervention distribution of $z_i$ into expressions involving only observed data, as such post-intervention distribution with do-operations cannot be directly estimated due to unobserved confounders [4,5]. The proof mainly relies on the **independent mechanism assumption** of data, which assumes causal independence among attributes, and **a conditional independence property of representations**, which can be enforced by minimizing mode-based CMI as shown in the revised **Proposition 2**.
>
> Our proof is given in **Appendix B.4**, and relevant theories are provided in **Appendix C.2**. We sincerely appreciate your profound academic insights, which have been truly inspiring in improving the rigor of our theory!
>
>
> *Reference:*
>
> [4] Robustly disentangled causal mechanisms: Validating deep representations for interventional robustness, ICML, 2019.
>
> [5] Desiderata for representation learning: A causal perspective, arxiv, 2021.

---

> > ### Author Response · Authors · 2024-11-21
> > **Response to Reviewer dyTi (Part 3 of 3)**
> >
> > ### W2: Mode label estimation
> >
> > There seems to be some misunderstanding regarding mode label estimation. Allow us to clarify this point:
> >
> > * During the data generation process, some confounders might affect the attributes and modes through underlying mechanisms, resulting in correlated data.
> >
> > * We pre-train an encoder on the correlated data with supervised attribute prediction losses, and then perform clustering on the **representations** of $a_k$ to estimate the mode labels. Specifically, for each attribute value $\alpha$, we select all the samples under $a_k=\alpha$, then perform clustering on their representations $z_k$ extracted by the pre-trained encoder. Then the discovered modes under different values of $a_k$ are labeled together to formulate the categorical $m_k$.
> >
> > * Both pre-training and clustering are performed on **correlated data**, thus the effects of causal mechanisms in data generation are unchanged. In particular, the samples for clustering under $a_k=\alpha$ exhibit hidden correlations between the true modes under $a_k=\alpha$ and the other attributes $a_{-k}$, which is caused by the confounders that affect $m_k$ and $a_{-k}$ while not affecting $a_k$ as you mentioned.
> >
> >
> > ### W3: Choice of attributes and modes in real-world datasets
> >
> > Thanks for bringing this up. As suggested, revisions have been made in **"Datasets", Section 5.1** to enhance clarity. For more details, please refer to **Appendix F**. For your convenience, we provide the explanations as follows.
> >
> > * CMNIST: **Parity check** identifies whether a digit is even or odd with multiple digits under each parity value. Accordingly, $a_1$ represents parity, $m_1$ represents the digits, and $a_2$ represents the color of digits, which is often correlated with digits, e.g., a player's jersey number may be associated with a specific color in sports.
> >
> > * UCI-HAR, RealWorld, and HHAR: **Wearable human activity recognition** identifies activities with variations under each activity. Accordingly, $a_1$ represents activity (e.g., "walking"), $m_1$ represents unknown activity modes (e.g., walking mode "stroll"), and $a_2$ represents user ID, which is often correlated with activity due to personal behavior patterns.
> >
> > * MFD: **Machine Fault diagnosis** identifies machine fault types with variations under each fault type, e.g., different forms of damages, pitting or indentation. Accordingly, $a_1$ represents fault type, $m_1$ represents unknown modes of fault types, and $a_2$ represents operating conditions, which could be correlated with machine faults.
> >
> > ### W4: Typos
> >
> > Thanks for your attention to detail. As suggested, we have made revisions to our paper and checked the paper thoroughly.
> >
> > Your feedback has been immensely valuable. We hope these explanations can address your concerns and earn a more favorable perspective on our submission. Should there be any additional concerns, please do not hesitate to contact us!

---

> ### Comment · Reviewer_dyTi · 2024-11-27
>
> I thank the authors for their response. However, I still have concerns regarding the theoretical aspect of the paper. In particular, while the introduction of the additional notation $z^l$ helps clarify some definitions, Definition 2 remains confusing because there is no mention of the node $z$ in the causal graph.
>
> Furthermore, I noticed that the authors changed the definition of Disentangled Representation (Definition 2) during the rebuttal phase. This change appears to accommodate a new result presented in Proposition 3. Although the proof of Proposition 3 seems correct based on my quick review, this alteration raises concerns about the rigor of the paper. Definition 2 is a critical foundation of the paper, and the authors have cited the same reference ([Wang & Jordan, 2021]) to introduce this definition both before and during the rebuttal phase. Given these issues, I will keep my score unchanged.

---

> ### Author Response · Authors · 2024-11-28
>
> We sincerely thank you for your feedback. We are glad to hear that our response clarifies the definitions of $z^l$, and deeply appreciate your effort to check the proof of Proposition 3. Your scientific rigor has inspired us.
>
> Allow us to further address your concerns about Definition 2 regarding the causal graph and the adjustment on Equation 2.
>
> ### **Causal graph with the extracted $z$**
> In our last revised version, we include such graph in the proof of Proposition 3, **Appendix B.4**. As agreed in W1.1, there is an edge from data $x$ to the extracted representations $z$, which is also supported by **Figure 2 in [2]**.
>
> In light of your suggestion, we agree that this graph is important, and **will move it to Section 3**. However, we might not be able to upload the revised version in a limited time, as it requires careful adjustment.
>
> ### **Adjustments on Equation 2**
> Thank you for bringing this up. Indeed, Definition 2 is the foundation of our paper. We have **shared** our adjustments on **Equation 2** in the response and revised paper, and welcome an open discussion.
>
> As mentioned in our previous response, "Definition 2 is optimized by omitting data ($x$) and attribute values ($\alpha_{-i}$) (in Equation 2)". This change is **part of the thorough adjustment to respond to W1.1**, which involves a new definition of extracted $z$ that differentiates from the true latent $z^l$ and affects Equation 2. We elaborate as follows:
>
> * Originally, we formulate Equation 2 with reference to [1] ([Wang & Jordan, 2021] as you mentioned), which might be misleading in **not differentiating** the extracted representations, the true latent representations, and the generative factors of data (i.e., attributes). This is fine in their work, as they mainly focus on the properties of the true latent.
>
>     Specifically, Equation 5 in [1] defines representations $Z$ as disentangled if, for $j=1, ..., d$ and any value $z_{-j}$ for $Z_{-j}$,
>
>     $P(Z_j|do(Z_{-j}=z_{-j}))=P(Z_j)$
>
>     where their $Z$ sometimes represents the extracted representations judging by context, but also represents the factors that generate data in Definition 3 [1], and the properties of this $Z$ are derived as the necessary condition for disentanglement in Theorem 4 [1], treating $Z$ as the true latent.
>
>     With reference to [1], although we did not explicitly differentiate true $z^l$ and extracted $z$ in the original version, we tried to imply that the $z$ in Equation 2 is the extracted representation by adding $x$ as the condition of $z$. We also explicitly differentiated $z$ from data attributes. Yet it was still unclear.
>
> [1] Desiderata for representation learning: A causal perspective, arxiv, 2021.
>
> * As you insightfully point out in W1.1, **the true and extracted representations need to be explicitly differentiated**, with which we entirely concur. During our revision, we found that this issue also affects **Equation 2, which needs to be modified according to our new definitions of $z$ and $z^l$**. To this end, we revised Equation 2 according to another reference [2].
>
>     Specifically, [2] uses $G$ to represent generative factors of data (i.e., attributes), and uses $Z$ to **explicitly represent the extracted representations**. They define the interventional effect of a group of generative factors $G_J$ on the extracted $Z$ as:
>
>     $p(z_I|do(G_J \gets \overset{\triangle}{g_J}))$
>
>     where $I, J$ denotes indices of representations and generative factors, and $z_I, \overset{\triangle}{g_J}$ represent some values of $Z_I, G_J$. This formulation is based on true generative factors (attributes) and the extracted representations, which fit our revised meaning of $z$. Thereby, we adopt this definition of interventional effect and formulate Equation 2 as:
>
>     $p(z_i|do(a_{-i}))=p(z_i)$
>
>     where attribute values ($a_{-i}=\alpha_{-i}$) is omitted for simplicity, and the right-hand side is formulated without do-operation, meaning $z$ is unaffected by such intervention.
>
> [2] Robustly disentangled causal mechanisms: Validating deep representations for interventional robustness, ICML, 2019.
>
> We believe this equation clearly reflects the interventional effect of attributes on the extracted representations. Then, we developed Proposition 3 based on it. **As a side benefit of your insightful comment W1.1, this revision absolutely paved the way for the link to disentanglement**, which relies on a clear definition of disentanglement.
>
> Finally, we assure you that **this change does not affect any other proofs or theories**. As you insightfully noticed, before Proposition 3, our theory did not directly link to Definition 2. Moreover, **the meaning of disentanglement never changed**, we only changed Equation 2 according to the refined meaning of $z$.
>
> We'd like to express our gratitude again for requesting clarification. We hope this addresses your concern, and would gladly answer any further questions.

---

### Official Review · Reviewer_UiJJ · 2024-11-05

**Soundness:** 3
**Presentation:** 2
**Contribution:** 2
**Rating:** 6
**Confidence:** 3

**Summary:**

The paper introduces a novel supervised disentanglement method, SD-HC, designed to address hidden correlations in multi-modal data, which traditional DRL methods often overlook. SD-HC focuses on preserving essential mode information within attributes by minimizing mode-based conditional mutual information, thus achieving better disentanglement and predictive accuracy under both hidden and attribute correlations. The authors theoretically prove that mode-based conditional mutual information minimization is both necessary and sufficient for effective disentanglement in complex, correlated datasets.

**Strengths:**

- This paper establishes a novel theory for supervised disentanglement with hidden correlations. It consists of both necessary and sufficient conditions, serving as a significant improvememt for existing works.
- The experiments are comprehensive and the results are good enough.

**Weaknesses:**

- The motivation of the problem setting is not clearly discussed. In Figure 2, to study the hidden correlations among attributions, why not directly study $a_k$ and $a_{-k}$? The role or advantage of introducing the variable $m_k$ is not clear for me.
- It seems that the performance of SD-HC heavily relies on the mode label estimation. First, in real applications, how to determine the number of modes $N_m$? In addition, for complex data pattern, the data representations are not high-quality and the clutering is not accurate accordingly. In such case, is there any techniques to guarantee good performance?
- There are some ambiguous concepts in this paper. For example, 1) "Data mode". I do not find its concrete definition. This term is not common in machine learning and its meaning varies with contexts. 2) "Multi-modal". It seems not to refer to the commonly used meaning, such as text and image. More strict definitions is required to make this paper more clear.
- The writting needs to be further improved. For example, in line 155, $c^a$ and $c^m$ should be boldface.

**Questions:**

- From Section 3 and 4, only one attribution mode is considered. For more complex or practical scenarios, all attributions may have their mode variables. Whether the SD-HC can efficiently and effectively address them?

---

> ### Author Response · Authors · 2024-11-21
> **Response to Reviewer UiJJ (Part 1 of 2)**
>
> Dear Reviewer UiJJ:
>
> Thank you for your careful review and invaluable comments. Your affirmation of our work's theoretical novelty and experimental quality inspires us to further improve our work. More importantly, we deeply appreciate your detailed feedback and constructive suggestions. Accordingly, we have uploaded the latest revised version, with **blue** markings indicating new or modified content.
>
> For a better understanding, please allow us to first clarify the key concepts in W3, then address the other concerns followingly.
>
> ### W3: Concepts of "data mode" and "multi-modal"
>
> We appreciate your attention to the key concepts and we'd like to provide the following clarifications. You are right that our definitions of "multi-modal" differ from those referring to different data modalities such as text and image. Instead, we refer to the complex distributions of data. A formal definition is provided as follows, which is incorporated in **"Multi-Modality and Hidden Correlations", Section 3.1** to enhance clarity.
>
> * By **multi-modal** data distributions, we refer to complex data distributions that can be formulated as probabilistic mixture models [1], e.g., Gaussian mixture models $p(x)=\sum_{k=1}^{K} \pi_k N(x;\mu_k, \psi_k^2)$ with $K$ Gaussian components, where $\pi_k, \mu_k, \psi_k$ denote the weight, mean vector, and covariance matrix of each component. Samples from such distributions might originate from different components of the mixture models, concentrating around different centroids and forming different clusters.
>
> * **Data modes** correspond to the components in the mixture model. Each sample originates from one component, which is indicated by its **mode label**. Kindly note the distinction between data modes (a property of data) and mode labels (the labels assigned to each sample based on its mode).
>
> *Reference:*
>
> [1] Minimax theory for high-dimensional Gaussian mixtures with sparse mean separation, NeurIPS, 2013.
>
> ### W1: Motivation for introducing $m_k$
>
> Regarding "hidden correlations", it seems there may be some misunderstanding. Allow us to clarify our problem setting and the different correlation types first, then explain the advantage of introducing $m_k$ with an intuitive example.
>
> * **Problem setting and difference between attribute correlations and hidden correlations**: Following the common setting of supervised disentangled representation learning (DRL) [2], we assume the attribute labels are known. In this case, **attribute correlations** (between $a_k$ and $a_{-k}$ as you mentioned) are **observable**, which makes DRL under attribute correlations a more tractable problem. We assume data follow multi-modal distribution under each value of $a_k$, where **data modes are unknown**. Accordingly, **hidden correlations** are defined as the correlation between the data modes under $a_k$ and other attributes $a_{-k}$, which are **unobservable** with modes unknown. This is a realistic yet challenging problem that deals with underlying variations and correlations related to supervised attributes.
>
> * **Advantage of introducing $m_k$**:
> The introduction of the mode variable $m_k$ is crucial for discovering and preserving the inherent structure of the complex data distribution under attribute $a_k$. Data modes capture the underlying variations when attribute values are held constant, and mode information can be important for attribute prediction [3]. Without explicitly considering modes, we risk losing important information for predicting attribute $a_k$.
>
> * **Intuitive example**: For the activity attribute in human activity recognition, "walking" activity can involve different modes like casual "stroll" or energetic "skip walk". Without encoding these modes, a model might confuse walking modes with other similar activities, e.g., confusing "skip walk" with "climbing down" due to their similar leaping motions. Encoding modes allow the model to explicitly recognize different walking modes as "walking" activities and better differentiate them from other activities. A case study on human activity recognition has been performed in **"WHAR Visualizations", Section 5.4**.
>
> In addition, we'd like to point out that our proposed method can deal with both attribute correlations between $a_k$ and $a_{-k}$ and hidden correlations between $m_k$ and $a_{-k}$, thus being **inclusive rather than exclusive of attribute correlations**. For theoretical and experimental details, please refer to **Section 3.3** and CMNIST settings in **"Evaluation Protocols", Section 5.1**.
>
>
> *Reference:*
>
> [2] Disentanglement and generalization under correlation shifts, CoLLAs, 2022.
>
> [3] Adaptive local linear discriminant analysis, TKDD, 2020.

---

> ### Author Response · Authors · 2024-11-21
> **Response to Reviewer UiJJ (Part 2 of 2)**
>
> ### W2: Mode label estimation
>
> You are right that *the performance relies on mode label estimation*. We highlight the analysis we have performed on this matter, and offer more practical guidance as follows:
>
> (1) **The number of modes**
>
> * We choose the number of modes $N_m$ by **hyper-parameter tuning**, as stated in **"Mode Label Estimation", Section 4**. We analyze the parameter sensitivity in **"Parameter Sensitivity of $N_m$", Appendix I**, and observe that SD-HC is **robust** to $N_m$ when it's equal to or slightly above the ground-truth number of modes. This is mentioned in **"Additional Analysis", Section 5.4**.
>
> * Alternatively, we could **estimate the number of modes in a data-driven manner**. This reduces the computational load of hyper-parameter tuning, but requires **expert knowledge** to choose the suitable method: For well-separated clusters, Elbow Method [4] would be suitable for estimating $N_m$ with k-means clustering. For complex and overlapping clusters, Bayesian Information Criterion [5] would be suitable with Gaussian Mixture Models for clustering. As suggested, we have added discussions in **"Parameter Sensitivity of $N_m$", Appendix I** for practical guidance and mentioned this in **"Mode Label Estimation", Section 4**.
>
> (2) **Challenging clustering on complex data**
>
> * We have evaluated **clustering performance** under varying noises in **"Clustering Evaluation", Appendix I**, with the results in Figure 15ab. We find that the **clustering performance drops as noise level increases**, because it is harder to acquire high-quality representations on complex data with large noises, as you mentioned.
>
> * We have analyzed **attribute prediction performance** under varying noises in **"The Impact of Noise Level and Train Hidden Correlation", Section 5.4**, with the results in Figure 8ab. We find that as noise level increases: (1) the gap between SD-HC (based on estimated mode labels) and SD-HC-T (based on **ground-truth mode labels**) slightly increases then stabilizes; and (2) the performances of all methods drop, yet SD-HC remains superior in general. This indicates that **although SD-HC's performance is slightly affected by poor clustering on noisy data, it still surpasses the baselines due to partial preservation of mode information**.
>
> * In addition, **deep clustering** could be explored for complex data. Deep clustering methods adopt alternative representation learning and clustering, so that the two tasks can mutually enhance each other, and sometimes incorporate self-supervised learning to improve the quality of representations [6,7]. As per your suggestion, we add discussions in **"Clustering Evaluation", Appendix I** for practical guidance and mentioned this in **"Mode Label Estimation", Section 4**.
>
> *Reference:*
>
> [4] The determination of cluster number at k-Mean using Elbow method and purity evaluation on headline news, iSemantic, 2018.
>
> [5] A widely applicable Bayesian information criterion, JMLR, 2013.
>
> [6] Deepdpm: Deep clustering with an unknown number of clusters, CVPR 2022.
>
> [7] Self-supervised convolutional subspace clustering network, CVPR 2019.
>
> ### W4: Typo
>
> Thanks for your scientific rigor. As suggested, we have checked the paper and made revisions.
>
> ### Q1: Multiple attributes with multi-modality
>
> A quick answer is **YES**, SD-HC can address the complex scenario where any number of attributes exhibit multi-modality. We provide theoretical and implementational discussions to support this statement.
>
> * **Theoretical guarantees**: As mentioned in **"Theoretical Contributions", Section 3.3**, our theoretical results focus on disentangling **one arbitrary** attribute $a_k$ among $K$ attributes, and show that one independence constraint is sufficient to disentangle $a_k$. For $a_k$ with multi-modality, this constraint minimizes conditional mutual information based on $m_k$. Similarly, for another $a_j, j \ne k$ with multi-modality, we can disentangle it the same way as $a_k$, i.e., we estimate its mode labels $m_j$ and add one independence constraint based on $m_j$ between representations $z_j$ and other representations $z_{-j}$.
> * **Implementational flexibility**: As mentioned in the second paragraph of **"Framework", Section 4**, our framework can be tailored to disentangle multiple attributes with multimodality. The specific adjustments involve constructing a discriminator and mode predictor for this attribute, and adding mode prediction loss and independence constraint based on its estimated mode labels.
>
>
> Thanks again for your invaluable comments. We hope our responses address your concerns and earn a more favorable perspective on our submission. Should there be further questions, we'd love to answer them!

---

> > ### Comment · Reviewer_UiJJ · 2024-11-25
> >
> > Dear authors,
> >
> > Thank you for your clarification. I have known the meanings of these terms. Please merge them into your draft for better understanding. For other questions, your responses have addressed most of my concerns. I am willing to raise my score to 6.

---

> > > ### Author Response · Authors · 2024-11-26
> > >
> > > We are immensely grateful for your recognition of our work and are deeply honored by your decision to increase the score!
> > >
> > > Guided by your suggestions, we have reflected on our work and have made revisions to incorporate or highlight the points discussed, including **clarifications on key concepts**, analysis and discussions on clustering, and the handling of multiple attributes with multi-modality, which have further improved the clarity and rigor of our work.
> > >
> > > Rest assured that we will continue to revise the paper to ensure that the added content is more prominent and clearly presented. In the meantime, if there are any aspects that could be further enhanced, we'd be thrilled to hear from you again and make the effort to further improve our work.
> > >
> > > Best regards

---

> ### Author Response · Authors · 2024-12-01
> **A supplementary note for Reviewer UiJJ**
>
> Thank you again for reviewing our paper and raising the score! Over the past few days, we have been diligently studying your comments, gaining a deeper understanding of the issues raised. Although it appears that you require no more clarifications, we'd like to provide additional results in response to your comment **Q1 (Multiple attributes with multi-modality)**.
>
> Specifically, we experiment on a new toy dataset as described below, and have validated the effectiveness of SD-HC in a more complex scenario.
>
> ### Data construction
>
> The new toy data are 4-dimensional with **4 attributes** ($a_i, 1 \le i \le 4$), where 2 attributes ($a_1, a_2$) exhibit multi-modality. We do not introduce multi-modality for all attributes, so that the impact on the attributes **without multi-modality** can still be observed.
>
> The new toy dataset is an **extension of the original toy dataset** in our paper. Similar to the original toy data, each data axis is controlled by one attribute, i.e., $x_i$ is affected by $a_i$ and noises, and unaffected by other attributes $a_{-i}$. Here $a_1$ and $a_2$ with multi-modality are constructed the same as the $a_1$ in the original toy data, with 3 modes under each attribute value; $a_3$ and $a_4$ are constructed the same as the $a_2$ in the original toy data. The mappings from attribute/mode labels to the corresponding data axis are also the same.
>
> ### Experiment settings
>
> We use the same settings as the original toy data, where we train on correlated data, and evaluate on three test sets, namely test 1 with the same correlations, test 2 without correlations, and test 3 with anticorrelations. **Complex attribute and hidden correlations** are introduced on the train data, e.g., $I(a_2;a_4) = 0.07, I(a_3; a_4)=0.13, I(m_1;a_2|a_1)=0.14, I(m_1;a_4|a_1)=0.36, I(m_2;a_3|a_2)=0.28$, where mutual information $I(\cdot;\cdot)$ measures the correlations between variables.
>
> The task is to learn disentangled representations for each attribute. We compare SD-HC with BASE (with supervised losses only), A-CMI, and SD-HC-T (with ground-truth mode labels). For SD-HC(-T), we use mode-based conditional mutual information (CMI) minimization for $a_1$ and $a_2$ with multi-modality, and attribute-based CMI minimization for $a_3$ and $a_4$. For A-CMI, attribute-based CMI minimization is used for all attributes.
>
> ### Results
>
> |BASE|$a_1$|$a_2$|$a_3$|$a_4$|
> |-|-|-|-|-|
> |test 1|0.965|0.871|0.999|0.999|
> |test 2|0.757|0.697|0.997|0.998|
> |test 3|0.539|0.585|0.997|0.994|
>
> |A-CMI|$a_1$|$a_2$|$a_3$|$a_4$|
> |-|-|-|-|-|
> |test 1|0.510|0.593|0.998|0.525|
> |test 2|0.507|0.537|0.997|0.507|
> |test 3|0.507|0.518|0.996|0.476|
>
> |SD-HC|$a_1$|$a_2$|$a_3$|$a_4$|
> |-|-|-|-|-|
> |test 1|0.994|0.797|0.998|0.999|
> |test 2|0.870|0.740|0.998|1.000|
> |test 3|0.752|0.741|0.997|0.999|
>
> |SD-HC-T|$a_1$|$a_2$|$a_3$|$a_4$|
> |-|-|-|-|-|
> |test 1|0.999|0.997|0.997|1.000|
> |test 2|0.966|0.982|0.996|1.000|
> |test 3|0.928|0.964|0.998|0.999|
>
> The attribute prediction accuracy is reported above, which shows that:
> * **A-CMI** performs badly on $a_4$, even though $a_3, a_4$ do not exhibit multi-modality and are easy to predict for the BASE method. This is because under hidden correlations $I(m_1;a_4|a_1)$, attribute-based CMI minimization on $a_1$ might also hurt the predictive ability of the representations of $a_4$, as proved in Proposition 1.
> * **SD-HC-T** outperforms SD-HC by a larger margin compared to the original toy data, probably because:
> (1) The increased complexity of data hinders clustering for estimating $m_i$, introducing errors via the corresponding mode-based CMI and mode prediction losses based on $m_i$;
> (2) The disentanglement of one representation also relies on the *Informativeness* of other representations, as shown in Proposition 2. Therefore, the disentanglement for $a_1$ might also be affected by mode label estimation errors on the other attribute $a_2$, which causes the loss of mode information and hurts the *Informativeness* of $z_2$, and vice versa.
> * **SD-HC** generally shows **superiority** compared to BASE and A-CMI. For attributes $a_1,a_2$ with multi-modality, SD-HC estimates mode labels and minimizes mode-based CMI, which can preserve their mode information to some degree. For simple attributes $a_3, a_4$, SD-HC considers their hidden correlations with the modes of other attributes, which can preserve their predictive abilities.
>
> We hope this empirical evidence further validates the effectiveness of SD-HC under more complex multi-modality, along with the theoretical and implementational aspects in our previous response. Rest assured that these additional results will be incorporated into our paper. As the discussion ends soon, we express our gratitude for your valuable comments and would greatly appreciate your continued support. Thank you very much for your time and thoughtful consideration.

---

### Author Response · Authors · 2024-11-22
**Summary of Revisions and Global Response**

We sincerely thank all the reviewers for their insightful reviews and valuable comments, which are instructive for us to improve the clarity and rigor of our paper.

**Towards handling hidden correlations under certain attributes in disentangled representation learning**, our paper proposes Supervised Disentanglement under Hidden Correlations (SD-HC), which discovers data modes under certain attributes and minimizes mode-based conditional mutual information to achieve disentanglement. SD-HC is equipped with both **solid theoretical guarantees from the perspective of mutual information and causality**, and brings **promising experimental results on synthetic data and real-world image and time series data**. In addition, SD-HC can be **flexibly** tailored when different numbers of attributes need to be disentangled, and applies to **multiple attributes, various correlation types, and simple uncorrelated and uni-modal cases**.

We are delighted that the reviewers generally held positive opinions of our paper, in that the investigated problem is "**challenging**", "**open**", "**important**" and "**prevalent in real-world data**" (Reviewer dyTi, wrRV, and HxXe), the proposed method and theory are "**properly defined**", "**clearly stated**", "**novel**", "**serving as a significant improvement for existing works**" (Reviewer UiJJ and HxXe),  the empirical evaluation is "**thorough**","**sufficient**", "**extensive**" and "**promising**" (Reviewer UiJJ, dyTi, wrRV, and HxXe).

The reviewers also raised insightful and constructive concerns. We have made every effort to address all the concerns by providing detailed clarifications, sufficient evidence, and in-depth analysis. We have uploaded the latest revised version, with **blue** markings indicating new or modified content. Here is the summary of the major revisions:

* **Enhance the rigor, completeness, and clarity of our theoretical results (Reviewers UiJJ, dyTi, wrRV)**: Our main theoretical results and contributions remain unchanged. We further enhance the proposed theory by adding definitions, differentiating true and learned representations, highlighting assumptions, and adding one more proof for rigor, completeness, and clarity throughout Section 3.
* **Better connect our theoretical results with practice (Reviewer UiJJ, dyTi, HxXe)**: We highlight the connections between learning and theory in "Theoretical Contributions", Section 3.3, and add generalization discussions to simple cases in "Generalization to Multiple Attributes and Simple Cases", Section 3.3, with more discussions added in Appendix B.5. In addition, we add toy experiments with multiple attributes and complex multi-modality and correlations to validate our method in complex scenarios, which will be added to our paper. These clarify how to implement our learning theory, and in what scenarios are they applicable.
* **Add practical guidance on the choice of clustering methods and mode estimation methods (Reviewer UiJJ)**: As the clustering method is a choice of design in our pre-training stage, we add guidance on how to choose appropriate clustering and mode estimation methods in "Clustering Evaluation" and "Parameter Sensitivity of $N_m$", Appendix I as mentioned in "Mode Label Estimation", Section 4.
* **Add the reasons for the settings on real-world data (Reviewer dyTi)**: We add explanations on the real-world phenomena that support our choices of attributes and modes on each real-world dataset.
* **Add coherent examples (Reviewer wrRV)**: We add examples in "Intuitive Example", Section 3.1. Along with the explanations on real-world datasets, the coherent examples in the same context of activity recognition are more complete and aid the understanding of our method.
* **Correcting a few typos and refining some expressions (Reviewer dyTi, HxXe)**: We check our paper thoroughly and correct a few typos regarding the symbols and figure references. Also, we refined some expressions to be more concise and save space for the added content.

The valuable suggestions from the four reviewers are highly valued. **We eagerly await your feedback and will gladly answer further questions.**

Best Regards!

---

### Author Response · Authors · 2024-12-03
**Summary of discussion**

We would like to provide a summary of the discussion phase to offer greater clarity.

The four reviewers generally recognized the strengths of SD-HC as follows:

* **Importance of the studied problem**. **Reviewers dyTi, wrRV, and HxXe** agreed that SD-HC studies an **important and realistic** problem for disentangled representation learning, i.e., the disentanglement of attributes with multi-modality and hidden correlations.
* **Theoretical contributions**. Although **Reviewer UiJJ** raised some doubts about our problem setting, Reviewer UiJJ recognized our theoretical results as "novel" and "a significant improvement for existing works". Specifically, based on conditional mutual information, we give the *necessary and sufficient conditions* for disentanglement under hidden correlations and attribute correlations, which is also the **first** sufficient condition for disentanglement under correlations. Our results can be generalized to various cases and provide theoretical foundations for various disentangling methods with conditional independence.
* **Empirical soundness**. **ALL reviewers** agreed that our comprehensive experiments on toy data and real-world *image and time series* datasets provide sufficient evidence to support the advantage of SD-HC. In addition to the manually introduced correlations as commonly studied, we experiment under **natural hidden correlations** within real-world data, **bringing our theoretical results into diverse real applications**. Extensive analyses are carried out to study the behaviors of different methods, the impact of various factors, and the learned representation distributions and decision boundaries.

While the reviewers recognized the above strengths of SD-HC, they also expressed some concerns:

* **Reviewer UiJJ** felt that the problem setting and mode-related concepts need to be further clarified, and also asked about the effectiveness of SD-HC under complex scenarios when mode label estimation is challenging and when multiple attributes exhibit multi-modality.

* **Reviewer dyTi** mainly required more theoretical explanations, including the meaning of representations $z$ in Figure 3, the assumptions, and the link between our theoretical results to disentanglement, and also required explanations regarding the mode label estimation step and experiment settings on real-world datasets.

* **Reviewer wrRV** raised similar concerns as Reviewer dyTi regarding the meaning of $z$ in our causal graphs and the link to disentanglement, and raised similar concerns as Reviewer UiJJ about the effectiveness of SD-HC under potential mode label estimation errors. More coherent examples were also required to aid readability.

* **Reviewer HxXe** felt that the connections between theory and learning could be better established, and the theoretical results could be more self-contained.

During the discussion phase, we diligently and thoroughly addressed each reviewer's concerns. The revisions are summarized in our **global response**.

* **Theoretically**, we added formal definitions of mode, differentiated true and learned representations $z$, highlighted the assumptions, and added one more proof that links to disentanglement. We also clarified the connection between our theory and learning and refined our results to be more self-contained.

* **Empirically**, we highlighted the analyses regarding the clustering performance for mode label estimation and the impact on the performance of SD-HC, which had been a part of our comprehensive experiments but might not be prominent to readers. We also provided practical guidance for choosing suitable clustering methods. In addition, we add toy experiments to validate SD-HC in a complex multi-attribute setting.

Eventually:
* **Reviewer UiJJ** acknowledged the understanding of the concepts, and raised the rating.
* **Reviewer HxXe** appreciated our clarifications, considered the other reviews/responses, and maintained a positive rating.
* **Reviewer dyTi** acknowledged that our new definitions $z, z^l$ were clear, and our added proof seemed correct, but raised new concerns. We further offered clarifications that the required graph was in the appendix and promised to move it to our main paper, and that the change of the equation in Definition 2 was part of the thorough adjustment regarding the new definition of $z$. Although we didn't receive further feedback, we sincerely hope this addresses the concerns.
* **Reviewer wrRV** might have been occupied during the discussion phase and was unable to engage in discussions.

We thank the reviewers for their effort and time in reviewing our paper, and the Area Chairs' effort in encouraging discussions. We hope this summary will assist all reviewers and the Area Chairs in making the final decision.

---

### Meta-Review · Area_Chair_8t62 · 2024-12-20

**Metareview:**

The paper aims to address disentangled representation learning under hidden correlations and multi-modal attribute distributions.
During the review, reviewers raised theoretical concerns, particularly regarding assumptions, definitional clarity, and the rigor behind the necessity and sufficiency claims. Reviewer dyTi and Reviewer wrRV still harbored concerns about theoretical rigor, the modifications of key definitions during the rebuttal, and the underlying assumptions needed to fully justify the claims.  No reviewers were willing to champion this paper.  we recommend rejecting this paper.

**Additional Comments On Reviewer Discussion:**

During the rebuttal period, the rigorous of major theoretical claims was raised by the reviewers:

- *Reviewer dyTi* and *Reviewer wrRV* expressed concerns about the clarity and stability of the theoretical foundations. Changes to a key definition (Definition 2) and the causal graph during the rebuttal phase raised questions about the paper’s rigor. The necessity and sufficiency claims appear to be strongly tied to specific assumptions that the reviewers felt needed more explicit detailing.

- After the causal graph was modified during the rebuttal, a new concern was raised. Specifically, *Reviewer wrRV* remained unconvinced about how certain paths would be blocked or how certain independence conditions could be established without additional parametric or structural assumptions. For instance, they asked, "How would you be able to render $z_1^l$ independent from $a_2$ given $x$, since the path $a_2 \to z_2^l \to x \to z_1^l$ exists?"

The AC has also carefully reviewed the original submission, the revised version, and all discussions. We believe these points are important and need to be clearly addressed, and a major revision is required. For the benefit of this paper, we reject it for now. Please note that this should not be taken as discouragement. We believe that this paper can be a strong submission after addressing these concerns.

---

> ### Public Comment · ~Rong_Hu4 · 2025-02-21
> **Response to the new concerns raised after the rebuttal period**
>
> We thank the area chair for the effort and support in reviewing our submission, and for sharing the new concern. We'd like to make clarifications about the new concern raised by Reviewer wrRV after the official rebuttal and discussion period, as Reviewer wrRV did not engage in discussions with us.
>
> In short, the independence conclusions are reached based on the structural assumptions in Definition 1, and no additional parametric assumptions are needed as explained below. Also, note the distinction between **causal independence** and **independence**, which seems to have caused some confusion.
>
> * **Data generation assumption in Definition 1**: Our model is built upon the data generation process of Definition 1, which includes the independent mechanism assumption that assumes attributes are **causally independent**. Note that this only requires that attributes are not the causal cause of one another, but **allows them to be correlated due to confounding**. Definition 1 includes structural assumptions about the causal graph of data generation.
>
> * **Blocking paths in Figure 3**: The independence in Figure 3 is derived using causal theorems about d-separation and backdoor paths, which only rely on the given causal graph structure based on our data generation process, and no additional parametric assumptions are needed.
>
> * **Independence in Proposition 2**: Proposition 2 addresses the problem that "if certain conditions are satisfied by representation learning, then what properties do the learned representations have?" The conditions "if ..." can be achieved by our learning objectives of minimizing supervised losses and conditional mutual information. Specifically, the independence $I(z_1; z_2|m_1)=0$ can be achieved using adversarial training as elaborated in "Losses", Section 4. The conclusions "then ..." are reached purely by **mutual information calculations**, and no additional structural and parametric assumptions are involved other than the given conditions in Proposition 2.
>
> * In addition, we **do not** render $z_1^l$ independent from $a_2$. In fact, they should be correlated if attribute correlations or hidden correlations exist between  $a_1, m_1$ and $a_2$. We're not sure where the confusion may arise. In line 195, we implied that they are **causally independent**, which means that $a_2$ is not the cause of $z_1^l$, as $z_1^l$ is the true latent representation of $z_1$.

---

### Decision · Program_Chairs · 2025-01-22

Reject